

# GLEAM v3: satellite-based land evaporation and root-zone soil moisture

Brecht Martens[1], Diego G. Miralles[2,1], Hans Lievens[1], Robin van der Schalie[3,2], Richard A.M. de Jeu[3], Diego Fernández-Prieto[4], Hylke E. Beck[5], Wouter A. Dorigo[6,1], and Niko E.C. Verhoest[1]

[1]Laboratory of Hydrology and Water Management, Ghent University, Coupure links 653, 9000 Ghent, Belgium
[2]Department of Earth Sciences, Vrije Universiteit Amsterdam, De Boelelaan 1085–1087, 1081 HV Amsterdam, Netherlands
[3]Transmissivity/VanderSat B.V., Space Technology Center, Huygensstraat 34, 2201 DK Noordwijk, Netherlands
[4]European Space Research Institute (ESRIN), European Space Agency (ESA), Via Galileo Galilei 64, 00044 Frascati, Italy
[5]Joint Research Centre (JRC), European Comission, Via Enrico Fermi 2749, 21027 Ispra, Italy
[6]Department of Geodesy and Geo-Information, Vienna University of Technology, Gußhausstraße 25–29, 1040 Vienna, Austria
*Correspondence to:* Brecht Martens (Brecht.Martens@ugent.be)

**Abstract.**

The Global Land Evaporation Amsterdam Model (GLEAM) is a set of algorithms dedicated to the estimation of terrestrial evaporation and root-zone soil moisture from satellite data. Ever since its development in 2011, the model has been regularly revised aiming at the optimal incorporation of new satellite-observed geophysical variables, and improving the representation
of physical processes. In this study, the next version of this model (v3) is presented. Key changes relative to the previous version include: (1) a revised formulation of the evaporative stress, (2) an optimized drainage algorithm, and (3) a new soil moisture data assimilation system. GLEAM v3 is used to produce three new data sets of terrestrial evaporation and root-zone soil moisture, including a 35-year data set spanning the period 1980–2014 (v3.0a, based on satellite-observed soil moisture, vegetation optical depth and snow water equivalents, reanalysis air temperature and radiation, and a multi-source precipitation
product), and two fully satellite-based data sets. The latter two share most of their forcing, except for the vegetation optical depth and soil moisture products, which are based on observations from different passive and active C- and L-band microwave sensors (European Space Agency Climate Change Initiative data sets) for the first data set (v3.0b, spanning the period 2003–2015) and observations from the Soil Moisture and Ocean Salinity satellite in the second data set (v3.0c, spanning the period 2011–2015). These three data sets are described in detail, compared against analogous data sets generated using the previous
version of GLEAM (v2), and validated against measurements from 64 eddy-covariance towers and 2338 soil moisture sensors across a broad range of ecosystems. Results indicate that the quality of the v3 soil moisture is consistently better than the one from v2: average correlations against in situ surface soil moisture measurements increase from 0.61 to 0.64 in case of the v3.0a data set and the representation of soil moisture in the second layer improves as well, with correlations increasing from 0.47 to 0.53. Similar improvements are observed for the two fully satellite-based data sets. Despite regional differences, the quality
of the evaporation fluxes remains overall similar as the one obtained using the previous version of GLEAM, with average correlations against eddy-covariance measurements between 0.78 and 0.80 for the three different data sets. These global data



sets of terrestrial evaporation and root-zone soil moisture are now openly available at www.GLEAM.eu and may be used for large-scale hydrological applications, climate studies and research on land–atmosphere feedbacks.

## 1 Introduction

Climate change alters the complex interplay between land and atmosphere, significantly impacting different processes in the global hydrological cycle (Huntington, 2006; Wild et al., 2008; Miralles et al., 2014b). Analysing these impacts requires long-term, observational and consistent data sets of essential hydrological variables, such as soil moisture, precipitation and terrestrial evaporation (or *"evapotranspiration"*). While most of these variables can be relatively easily observed at different spatial scales, the large-scale observation of terrestrial evaporation is hampered by the inability to sense this flux directly from satellites. Consequently, this crucial return flow of water from land into the atmosphere remains one of the most elusive and uncertain components of the global hydrological cycle (Dolman et al., 2014; Miralles et al., 2016b).

However, the climate community is becoming increasingly aware of the crucial role that terrestrial evaporation plays in the Earth system, acting as a link in hydrological and biogeochemical cycles, and being a driver of air humidity, cloud formation, temperature or precipitation (Seneviratne et al., 2010; Taylor et al., 2012; Miralles et al., 2012). Consequently, past decades have seen significant efforts to enhance our understanding of the global magnitude and variability of this flux. Some of these efforts have mainly concentrated on routinely measuring evaporation in the field (Wang and Dickinson, 2012), resulting in the increased availability of in situ observations from different climatic regions across the globe (Baldocchi et al., 2001; Jung et al., 2009). In addition, acknowledging the sparseness of current in situ networks and their inability to meet the spatio-temporal requirements for climatological studies, the potential of satellite remote sensing to understand terrestrial evaporation dynamics has been intensively explored. While nowadays evaporation still remains undetectable from space, several models that combine remotely-observable drivers of this flux (e.g. radiation, air temperature, soil moisture) have been developed in recent years (Wang and Dickinson, 2012; McCabe et al., 2016).

Existing algorithms share the overarching objective of producing consistent, long-term, and global data sets of terrestrial evaporation, but the methods and input data sets used in these models markedly differ (e.g., Mu et al., 2007; Fisher et al., 2008; Zhang et al., 2010; Miralles et al., 2011; Loew et al., 2015). Recently, McCabe et al. (2016), Michel et al. (2016) and Miralles et al. (2016a) evaluated the relative value of four of these evaporation models using standardized satellite- and in situ based forcing data sets. Results highlighted substantial differences in model performance, especially under conditions of water stress. In addition, these studies found pronounced deficiencies in the way evaporation is partitioned into its different components (i.e. transpiration, bare-soil evaporation, open-water evaporation, interception loss and sublimation). Miralles et al. (2016a) also highlighted the importance to advance the physical representation of evaporation in these simple models, and the need to incorporate new technological advances in remote sensing science.

The Global Land Evaporation Amsterdam Model, GLEAM (Miralles et al., 2011) is the only global evaporation model that is primarily driven by microwave remote sensing observations, and that uses satellite soil moisture as a constraint on potential evaporation rates. Additional key features of the approach are the independent and detailed modelling of forest interception loss



based on Gash's analytical model (Gash, 1979; Valente et al., 1997; Miralles et al., 2010), and the use of microwave vegetation optical depth (VOD) as a proxy for the vegetation water content (Liu et al., 2013) in the calculation of the evaporative stress (Miralles et al., 2011, 2014b; Martens et al., 2016). Evaporation and soil moisture data sets from GLEAM have been widely used in the past to study spatial variability and trends in the water cycle (e.g., Jasechko et al., 2013; Greve et al., 2014; Miralles

et al., 2014a; Zhang et al., 2016), as well as land–atmosphere feedbacks (e.g., Miralles et al., 2014b; Guillod et al., 2015). The first version (v1) of the model was developed in 2011 (Miralles et al., 2011) and further refined in 2014 (v2, Miralles et al. (2014b)); the present paper presents the third version (v3) of the methodology. In this new version, each of the key components of the methodology has been updated, except for the interception loss algorithm. First, aiming at a more realistic representation of evaporative stress, observations of microwave VOD and root-zone soil moisture have been optimally combined to represent

the non-linear response of soil and vegetation to the drying of land. Second, the soil module has been adapted to represent the continuous drainage of precipitation through the vertical profile. Finally, the soil moisture data assimilation system – recently updated and validated for Australia (Martens et al., 2016) – has been optimized to work at the global scale and to integrate different satellite soil moisture observations. These changes have respected the rationale of GLEAM of targeting only the fundamental processes controlling large-scale evaporation rates, while keeping the overall simplicity and observational nature

of the model.

The new v3 has been used to produce three new data sets of terrestrial evaporation and root-zone soil moisture: v3.0a, v3.0b and v3.0c. The v3.0a data set is a 35-year (spanning the period 1980–2014) global record based on satellite-based soil moisture, VOD and snow water equivalents, reanalysis air temperature and radiation, and a multi-source precipitation product. The v3.0b (spanning the period 2003–2015) and v3.0c (spanning the period 2011–2015) are fully satellite-based, quasi-global

(50°N-50°S) data sets, which share most of their forcing database, except for the VOD and soil moisture observations. In case of the v3.0b data set, these are based on a merge of observations from different passive and active C- and L-band microwave sensors (Liu et al., 2011, 2012; Wagner et al., 2012). The v3.0c data set on the other hand relies on observations of the Soil Moisture and Ocean Salinity (SMOS) satellite, the first L-band mission dedicated to the observation of surface soil moisture (Kerr et al., 2001).

The main goal of this study is to present the new version of GLEAM and the corresponding evaporation and root-zone soil moisture data sets, including a global validation using a large database of in situ soil moisture and evaporation measurements. In addition, the quality of these data sets is compared against analogous datasets generated using the former version of GLEAM, allowing to evaluate the added value of the new formulations. The paper is organized as follows: Sect. 2 describes the new algorithms, highlighting the main changes upon the previous version. The forcing data and the in situ validation data

are described in Sect. 3. Section 4 analyzes the quality of the GLEAM data sets and discusses the results, while the main conclusions are summarized in Sect. 5.



## 2 Methodology

### 2.1 Baseline description of GLEAM

GLEAM (see Fig. 1) is a set of algorithms designed to estimate evaporation from satellite observations. It separately derives the different components of terrestrial evaporation, i.e. transpiration, bare-soil evaporation, open-water evaporation, interception

loss and sublimation (Miralles et al., 2011). Each grid cell is considered to comprise four different land-cover types: (1) bare soil, (2) low vegetation (e.g. grass), (3) tall vegetation (e.g. trees) and (4) open water. These fractions are sourced from the Global Vegetation Continuous Fields product (MOD44B), based on observations from the Moderate Resolution Image Spectroradiometer (MODIS). The evaporative flux is calculated for each of these fractions separately and then aggregated to the scale of the pixel based on the fractional cover of each land-cover type. First, the Priestley and Taylor (1972) equation is

used to calculate the cover-dependent potential evaporation rate $E_{\mathrm{p}}$ (mm/day) based on air temperature and net radiation:

$$\lambda E_{\mathrm{p}} = \alpha \frac{\Delta}{\Delta + \psi} (R_{\mathrm{n}} - G) \tag{1}$$

where $\lambda$ (MJ kg$^{-1}$) is the latent heat of vaporization and $\Delta$ (kPa K$^{-1}$) is the slope of the saturated water vapour-temperature curve. Both variables can be estimated using empirical relationships to the air temperature (Henderson-Sellers, 1984; Maidment, 1993). $\psi$ (kPa K$^{-1}$) is the psychometric constant, $\alpha$ (dimensionless) is the Priestley and Taylor coefficient, $R_{\mathrm{n}}$ (W m$^{-2}$)

is the net radiation and $G$ (W m$^{-2}$) is the ground heat flux. $G$ is calculated as a constant fraction of $R_{\mathrm{n}}$ depending on the cover type (Miralles et al., 2011). For $\alpha$, a value of 1.26 has been reported by Priestley and Taylor (1972) for well-watered grasslands, and has been used in numerous studies for a variety of ecosystems. However, empirical studies have highlighted the more conservative nature of tree stomata, generally resulting in lower rates of potential evaporation in forested areas (Shuttleworth and Calder, 1979; Kelliher et al., 1993; Teuling et al., 2010). Therefore, the $\alpha$ for tall vegetation is defined after the

findings by McNaughton and Black (1973), Shuttleworth et al. (1984), Viswanadham et al. (1991), Diawara et al. (1991) and Eaton et al. (2001), that report an average value of 0.97 (with a 0.08 standard deviation over the different studies) for forests during precipitation-free periods (i.e. no rainfall interception) and no evaporative stress.

Estimates of $E_{\mathrm{p}}$ are converted into actual evaporation $E$ (mm/day) – i.e. transpiration or bare-soil evaporation depending on the land-cover type – using a cover-dependent, multiplicative stress factor $S$ (dimensionless) ranging from 1 to 0. $S$ is

calculated as a function of microwave VOD and root-zone soil moisture. The latter is calculated using a multi-layer water-balance algorithm considering net precipitation (precipitation minus interception loss) and snowmelt as inputs, and evaporation and drainage as outputs. The depth of the root zone is a function of the land-cover type and comprises three model layers for the fraction of tall vegetation (0–10, 10–100 and 100–250 cm), two for the fraction of low vegetation (0–10, 10–100 cm), and only one for the fraction of bare soil (0–10 cm). Forest interception loss is estimated independently using the analytical model

introduced by Gash (1979) and further refined by Valente et al. (1997), forced with precipitation and considering both the characteristics of precipitation and vegetation (Miralles et al., 2010). We refer to Miralles et al. (2011), Miralles et al. (2014b),



and Martens et al. (2016) for detailed descriptions of the four modules. In the next section, we focus on the changes relative to the previous model version (Miralles et al., 2014b).

## 2.2 Recent advances in GLEAM

### 2.2.1 Soil module

Figure 2 shows a schematic of the conceptual root zone for the fraction of tall vegetation in a pixel. Each soil layer is subdivided in three different compartments. The first zone (bottom) represents the water retained below wilting point $w_{\mathrm{wp}}$ (m³ m⁻³) which is not available for root uptake. For the bare-soil fraction, the residual soil moisture $w_{\mathrm{r}}$ (m³ m⁻³) is used instead. The second compartment of the layer is bounded by $w_{\mathrm{wp}}$ and the porosity of the soil matrix $w_{\mathrm{p}}$ (m³ m⁻³), and represents the maximum volume of water available for drainage and evaporation. Finally, the third compartment represents the solid phase

of the soil column and thus cannot hold any water. The soil properties used in GLEAM come from the database of Global Gridded Surfaces of Selected Soil Characteristics generated by the International Geosphere-Biosphere Programme Data and Information System (IGBP-DIS, Global Soil Data Task Group (2000)).

At every (daily) time step $i$, the state of any layer $l$ is characterized by its water content $w_i^{(l)}$ (m³ m⁻³), which is updated using:

$$w_i^{(l)} = w_{i-1}^{(l)} + \frac{\left( F_{\mathrm{s},i}^{(l-1)} + F_{\mathrm{f},i}^{(l-1)} - E_{i-1}^{(l)} - F_{\mathrm{s},i}^{(l)} \right) \Delta t}{\Delta z^{(l)}} \tag{2}$$

where $w_{i-1}^{(l)}$ is the volumetric soil moisture content of layer $l$ at the previous time step $(i-1)$, $F_{\mathrm{s},i}^{(l-1)}$ (mm/day) is the volume of water slowly draining into the layer (slow draining flux), $F_{\mathrm{f},i}^{(l-1)}$ (mm/day) is the volume of water directly reaching the layer (fast draining flux), $E_{i-1}^{(l)}$ (mm/day) is the evaporative flux from the previous day, $F_{\mathrm{s},i}^{(l)}$ (mm/day) is the slow drainage of water out of the reservoir, $\Delta t$ is the temporal resolution (one day) and $\Delta z^{(l)}$ (mm) is the depth of the layer. Note that for the first

layer ($l = 1$), only $F_{\mathrm{f},i}^{0}$ is considered as an input, as there is no draining layer on top.

In the previous model versions, the entire volume of net precipitation (i.e. precipitation minus interception loss, plus snowmelt) was first stored in the top layer, which subsequently drained to field capacity into the next soil layer (Miralles et al., 2011); the same process was used to calculate the vertical flow from the remaining layers. As a result, the soil moisture could not exceed field capacity, nor was drainage allowed to occur below that threshold. In GLEAM v3, net precipitation is

first partitioned between the different soil layers based on the relative saturation at the beginning of the daily time step, in order to estimate the fast draining flux $F_{\mathrm{f},i}^{(l)}$. Next, the volume of water that slowly drains to the next layer ($F_{\mathrm{s},i}^{(l)}$) is estimated using a simplified representation of Darcy's law, in which a fraction of the available water above wilting point is drained to the next layer based on (1) the relative saturation of each layer and (2) the difference in soil moisture content between both layers.

The rationale behind this simple drainage algorithm is that the downward flux of water is expected to increase if (1) the

relative soil moisture content is higher (physically resulting in increased hydraulic conductivities), and (2) the difference in soil moisture between source and sink is larger (arising in higher differences in soil-water potential). This empirical drainage



algorithm is preferred over well-known alternatives such as the Richards equation, Brooks-Corey (Brooks and Corey, 1964) or Clapp-Hornberger (Clapp and Hornberger, 1978), due to its simplicity and the fact that it does not require the use of additional largely-unconstrained ancillary data on soil properties at the global scale.

### 2.2.2 Data assimilation system

The original Kalman filter approach to assimilate microwave soil moisture observations into the first soil layer (Miralles et al., 2011) was replaced in favour of a simple Newtonian Nudging algorithm (Miralles et al., 2014b) in the v2, which was further optimized by Martens et al. (2016). This Newtonian Nudging scheme minimizes the computational demands and fits well within the concept of GLEAM of keeping the model as simple and observation-based as possible. While more complex algorithms like the ensemble Kalman Filter have also been applied in GLEAM, the added value has shown to be marginal

(Lievens et al., 2016). Therefore, in this new version, a similar approach to the one implemented by Martens et al. (2016) – in which the soil moisture content of the first layer is updated using a Newtonian Nudging scheme – is adopted:

$$w_i^{(1)+} = w_i^{(1)-} + K\gamma \left( \hat{w}_i^o - \hat{w}_i^{(1)-} \right)$$

$(3)$

where $w_i^{(1)+}$ is the a posteriori soil moisture state at the first model layer (i.e. after application of the data assimilation algorithm), $w_i^{(1)-}$ is the a priori soil moisture state at the same layer (i.e. before assimilation of the observed soil moisture), $K$

(dimensionless) is the nudging factor (a value of 1 is used to maximize the impact of the assimilation algorithm as in Martens et al. (2016)), $\gamma$ (dimensionless) is the quality factor, and $\hat{w}_i^o$ (m³ m⁻³) and $\hat{w}_i^{(1)-}$ (m³ m⁻³) are the observed and modelled soil moisture anomaly, respectively. The latter two represent deviations relative to the seasonal climatology of soil moisture – calculated similar to De Lannoy and Reichle (2016) and Lievens et al. (2016) – as opposed to Martens et al. (2016), in which absolute values of soil moisture were assimilated.

As most assimilation algorithms require bias-free observations in reference to the modelled states, a bias removal algorithm prior to, or during the assimilation step has to be applied. However, no standard procedure exist to correct these constant or seasonally varying biases (Lievens et al., 2015; De Lannoy and Reichle, 2016), thus the choice of the bias-removal algorithm remains to some degree subjective. As indicated by Martens et al. (2016), the use of a classical CDF-matching approach prior to the assimilation step clearly introduced seasonal biases in the GLEAM soil moisture and evaporative fluxes. As a result,

in GLEAM v3 soil moisture anomalies are assimilated instead, as this approach allows the correction of potential seasonal biases between the modelled and observed soil moisture states (De Lannoy and Reichle, 2016). A similar approach to that by De Lannoy and Reichle (2016) and Lievens et al. (2016) is followed to obtain the soil moisture anomalies. As a Triple Collocation Analysis (TCA, Scipal et al. (2008); Miralles et al. (2010); Gruber et al. (2016)) is applied here to obtain the observation and model errors, the anomaly time series of the observations are scaled towards the modelled soil moisture

anomalies using a linear regression model prior to the assimilation (Yilmaz and Crow, 2013). We note that for applying a TCA, a third independent data set of the same geophysical variable is required. For this purpose, soil moisture fields from the Noah model in the Global Land Data Assimilation System (GLDAS) (Rodell et al., 2004) are used. The three independent and



rescaled anomaly time series of surface soil moisture are used in the TCA to estimate both the model and observation errors on a yearly basis. The latter two are then adopted to calculate the quality factor ($\gamma$) as in Martens et al. (2016):

$$\gamma = \frac{\sigma_{\mathrm{mod}}^{(1)}}{\sigma_{\mathrm{mod}}^{(1)} + \sigma_{\mathrm{obs}}} \tag{4}$$

where $\sigma_{\mathrm{mod}}^{(1)}$ (m$^3$ m$^{-3}$) and $\sigma_{\mathrm{obs}}$ (m$^3$ m$^{-3}$) are the standard deviations of the random model and observation errors, respectively.

5      Finally, in contrast to the assimilation of soil moisture observations in all model layers in GLEAM v2 (Miralles et al., 2014b; Martens et al., 2016), only the first model layer is updated in the new version. The latter choice is motivated by the slower dynamics of the deeper model layer, which may be strongly perturbed when soil moisture observations are assimilated into these layers using this simple Newtonian Nudging scheme. The impact of the soil moisture update in this v3 is thus propagated towards deeper layers by drainage processes only, which ensures a smooth transition of water through the vertical profile.

## 2.2.3   Stress module

The limiting effects of environmental factors such as water availability or heat stress, and phenological constraints acting on evaporation, are generally combined in a single empirical stress factor accounting for the decrease in potential evaporation (Sellers et al., 2007). In GLEAM, a multiplicative stress factor $S$ ranging between 0 (maximum stress and thus no evaporation) and 1 (no stress and thus potential evaporation) is defined.

15      In the first version (Miralles et al., 2011), $S$ was parameterized separately for the fractions of tall and short vegetation using non-linear relationships between $S$ and the soil moisture of the wettest layer. To account for changes in vegetation phenology of the (more dynamic) short vegetation, the VOD was also used in the calculation of $S$ for this fraction. These functions were linearized in the second version and the VOD was also introduced for the calculation of the stress for the fraction of tall vegetation (Miralles et al., 2014b; Martens et al., 2016). However, based on experimental evidence, a non-linear response of $S$ to soil moisture is expected for most vegetation types (e.g., Colello et al., 1998; Serraj et al., 1999; Ronda et al., 2002; Combe et al., 2016). As a consequence, non-linear stress functions are re-introduced in GLEAM v3 for both tall and short vegetation:

$$S = \sqrt{\frac{\mathrm{VOD}}{\mathrm{VOD}_{\mathrm{max}}} \left(1 - \left(\frac{w_{\mathrm{c}} - w^{(w)}}{w_{\mathrm{c}} - w_{\mathrm{wp}}}\right)^2\right)} \tag{5}$$

where $\mathrm{VOD}_{\mathrm{max}}$ (dimensionless) is the maximum VOD for a specific pixel, $w_{\mathrm{c}}$ (m$^3$ m$^{-3}$) is the critical soil moisture and $w^{(w)}$ (m$^3$ m$^{-3}$) is the soil moisture content of the wettest layer, assuming that plants withdraw water from the zone in which it is more easily accessible. As soil moisture decreases, $S$ decreases (i.e. increased evaporative stress), since water becomes less easily available for the roots. As vegetation phenology is not explicitly modelled, the VOD – closely linked to the vegetation water content (Liu et al., 2013) – is used to account for the effect of (seasonal or occasional) phenological constraints on evaporation (e.g. leaf-out, fires, pests etc.), with decreasing VOD resulting in lower values for $S$ and thus higher evaporative stress. As seen from Eq. 5 and Fig. 3, the stress function is thus defined by both the soil moisture content in the wettest soil





layer and the VOD. If $w^{(w)}$ reaches $w_{\mathrm{wp}}$, Eq. 5 implies that the vegetation is incapable to retrieve water from the soil and $S$ equals zero (and so does transpiration). On the other hand, for soil moisture values exceeding $w_c$, it is assumed that the water availability is not a limiting factor (i.e. $S = 1$ and transpiration equals potential transpiration).

Figure 3 illustrates the shape of the stress function in Eq. 5 for a pixel dominated by a strong seasonality in VOD (left-hand side, e.g. a savannah) and a site with a limited variability in VOD (right-hand side, e.g. a tropical rain forest). For illustrative purposes only, it is assumed that the soil properties ($w_c$ and $w_{\mathrm{wp}}$) are the same for both sites. As can be seen, where the range in VOD is expected to be low given the absence of a marked seasonality, $S$ would mainly depend on soil moisture. Conversely, if a large seasonality in the VOD is present (see left-hand side figure), the VOD becomes relatively more important for the calculation of $S$.

Finally, for the bare-soil fraction, $S$ is linearly related to the soil moisture state using the critical and residual soil moisture content as upper and lower boundary conditions, respectively:

$$S = 1 - \frac{w_c - w^{(1)}}{w_c - w_r} \tag{6}$$

Since only the top layer is considered for the fraction of bare soil, $S$ is fully driven by surface soil moisture ($w^{(1)}$).

## 3 Data

### 3.1 Input data sets

Table 1 gives an overview of the selected forcing data sets for GLEAM v3. All input data sets have a daily resolution and have been linearly re-sampled from their original spatial resolution (see Table 1) to a common 0.25° global grid. Given the aim to extract all valuable information on terrestrial evaporation from existing satellite records, forcing data sets are preferentially derived from satellite observations. However, since a key application of the GLEAM data sets is to analyze the impact of climate change on terrestrial hydrology, we also explore the use of alternative forcing data sets (e.g. reanalysis data) to yield multi-decadal records of terrestrial evaporation and root-zone soil moisture.

Radiation inputs are based on measurements from the Clouds and Earth's Radiant Energy System (CERES) onboard Terra and Aqua (Wielicki, 1996), which are available globally from the year 2001 onwards on a 1° regular grid. Additionally, radiation fluxes from the current reanalysis of the European Centre for Medium-Range Weather Forecasts (ECMWF), ERA-Interim (Dee et al., 2011), are also processed. ERA-Interim data are available globally from 1979 to present, with a temporal resolution of 3 hours and a spatial resolution of approximately 0.75°.

For the precipitation forcing, the Tropical Rainfall Measurement Mission (TRMM) Multi-satellite Precipitation Analysis (TMPA) 3B42v7 product (Huffman et al., 2007) and the Multi-Source Weighted-Ensemble Precipitation (MSWEP) data set (Beck et al., 2016) are selected. The TMPA 3B42v7 data set combines measurements from several satellites and is bias corrected using ground-based measurements of precipitation. The product is available for the period 1998–2015 and covers 50°N–



50°S based on a 0.25° grid. MSWEP on the other hand is based on a merger of selected satellite-, reanalysis- and gauge-based products, and is available from 1979 until 2014 at a 0.25° spatial resolution.

Air temperatures are derived from measurements of the Atmospheric Infrared Sounder (AIRS, Aumann et al. (2003)), which are available from 2003 onwards on a global 1° regular grid. Air temperature estimates from ERA-Interim are also used in this study. As for the radiation, data are available globally from 1979 until near present at 3-hourly intervals.

To estimate sublimation, snow-water equivalents from the European Space Agency (ESA) GLOBSNOW product (Luojus et al., 2013) are used. This data set is mainly based on retrievals from the Scanning Multichannel Microwave Radiometer (SMMR), Special Sensor Microwave/Imager (SSM/I), and the Advanced Microwave Scanning Radiometer (AMSR-E), and is available from 1980 onwards. The GLOBSNOW product only covers the Northern hemisphere and is therefore merged with the National Snow and Ice Data Center (NSIDC) monthly snow-water equivalent climatology product (Armstrong et al., 2005) for the Southern hemisphere. The latter is also based on measurements from SSMR and SSM/I.

As discussed in Sect. 2.2.3, the phenological controls on transpiration are derived from observations of microwave VOD. Here, the 0.25° product from Liu et al. (2011) is used, which is based on retrievals from several passive microwave sensors using the Land Parameter Retrieval Model (LPRM, Owe et al. (2008)). The product is available at the global scale and spans the period 1980–2012. In order to cover the period 1980–2015, the product is merged with LPRM-based VOD retrievals from SMOS (van der Schalie et al., 2015, 2016) using a similar CDF-matching approach as in Liu et al. (2011). The resulting data set contains gaps due to the repeating cycle of the satellites, the requirement of non-frozen conditions for parameter retrieval, and the presence of Radio Frequency Interference (RFI). In order to obtain smooth and continuous time series, the VOD data set is gap-filled using a moving average filter with a 7-day window. Remaining gaps, generally occurring in winter time due to freezing temperatures and snow covers, are linearly interpolated between the last and next available retrieval. We note however that in periods for which the land is covered by snow, the VOD is not used as the entire evaporation flux is assumed to correspond to sublimation. Finally, if any gaps remain, these are gap-filled using nearest neighbour interpolation. It should be noted here that microwave sensors operating at different frequencies might be sensitive to diverse components of vegetation, varying at different time scales (e.g. Guglielmetti et al., 2007; Liu et al., 2011). Despite the CDF-matching, which corrects for differences in long-term statistics, the use of different microwave sensors might impact the temporal dynamics of the VOD data set used here.

Finally, for the assimilation of microwave soil moisture in GLEAM, the SMOS Level 3 soil moisture product (Jacquette et al., 2010) and the ESA Climate Change Initiative soil moisture (ESA CCI SM v2.3) data set (Liu et al., 2012; Wagner et al., 2012) have been selected. The latter is a blended product of soil moisture retrievals from several active and passive microwave sensors, available for the period 1978–2015 at the global scale. In addition, surface soil moisture fields from the Noah model in GLDAS (Rodell et al., 2004) are used as a third independent data set in the TCA (see Sect. 2.2.2). Despite fundamental differences between GLEAM and Noah, some degree of dependency between their soil moisture estimates might be present due to the presence of common precipitation observations embedded within MSWEP, TMPA 3B42 and the forcing of GLDAS Noah. However, the merging schemes used to produce the precipitation data sets are ultimately different (Rodell et al., 2004; Huffman et al., 2007; Beck et al., 2016). Such a dependency may penalize the satellite-based soil moisture in the TCA (Yilmaz





and Crow, 2014), which would result in a lower quality factor $\gamma$ (see Eq. 4) applied in the data assimilation system and, subsequently, in a more conservative soil moisture update.

As discussed in Sect. 2, GLEAM also requires several static data sets describing the soil properties, land cover and average rainfall climatology. For the land cover fractions, the Global Vegetation Continuous Fields product (MOD44B) – based on observations from MODIS – is selected (Hansen et al., 2005). The high-resolution product at 250 m is chosen here and is up-scaled to the required grid size of 0.25° (note that in previous model versions, the low resolution 0.25° product produced by the MODIS team was used directly). Soil properties such as wilting point, soil porosity, field capacity and critical soil moisture are derived from the database of Global Gridded Surfaces of Selected Soil Characteristics (Global Soil Data Task Group, 2000). Finally, as in Miralles et al. (2010), a monthly rainfall intensity climatology is inferred from the Combined Global Lightning Flash Rate Density monthly product (Mach et al., 2007) produced by the National Aeronautics and Space Administration (NASA) agency.

Using various combinations of the forcing data described in Table 1, three different data sets of terrestrial evaporation and root-zone soil moisture are produced using GLEAM v3 (see Table 2). The inputs of snow water equivalent, the third independent data set used in the TCA and the static fields are shared by all data sets and are therefore not listed in Table 2. The first GLEAM data set (hereafter referred to as v3.0a) is a 35-year data set (1980–2014) covering the entire globe and is based on satellite-observed soil moisture, vegetation optical depth and snow water equivalents, reanalysis air temperature and radiation, and the MSWEP datset for precipitation. Given the multi-decadal coverage of this data set, it is intended to foster climatological research. The remaining data sets (v3.0b and v3.0c) are fully satellite-based, and span a significantly shorter period. In addition, these data sets only cover 50°N–50°S due to the use of the satellite-based TMPA 3B42v7 product. The differences between both satellite-based data sets are the VOD and soil moisture forcing, which are retrieved from SMOS only in the v3.0c data set, and from multiple active and passive microwave sensors in the v3.0b data set. This also implies a different record length of 13 (2003–2015) and 5 years (2011–2015) for the v3.0b and v3.0c data sets, respectively.

## 3.2   Validation data sets

For validation purposes, in situ soil moisture and evaporation measurements from different global networks are processed. Soil moisture measurements are sourced from the database of the International Soil Moisture Network (ISMN, Dorigo et al. (2011, 2013)), whereas the FLUXNET 2015 synthesis data set (http://fluxnet.fluxdata.org/) is used to obtain the in situ measurements of evaporation (see Table A1 for an overview of the selected sites). Several studies have already highlighted the lack of closure in the energy balance at eddy-covariance sites and a consequential tendency to underestimate latent heat fluxes (Wilson et al., 2002). Therefore, the corrected measurements of latent heat flux using the Bowen-ratio are used here. All available measurements for 1980–2015 are considered for inclusion in the validation set. Measurements are masked using the quality flags provided in the corresponding data set archives and aggregated from their native temporal resolution (generally 30 minutes or 1 hour) to the required daily scale. For the evaporation data sets, only days with less than 25 % of missing data are processed. Next, the resulting daily time series are screened for extreme outliers and repetitive measurements. Soil moisture measurements are subsequently masked for snow and air temperatures below 0°C using the snow water equivalents from GLOBSNOW and





the ERA-Interim air temperature data set, respectively (see Table 1). As eddy-covariance measurements are generally less reliable during precipitation, rainy intervals are masked from the data sets of in situ evaporation. Finally, only sites with at least 365 daily measurements after masking are included in the validation data set. This yields a total of 64 quality-checked eddy-covariance sites (see Table A1) and a total of 2338 soil moisture sensors covering various ecosystems across the globe.

Note that the soil moisture sensors are installed at different depths below the soil surface and used to validate both the first (0–10 cm, 1119 sensors) and second model layer (10–100 cm, 1219 sensors), depending on the installation depth. Sensors located in the same GLEAM grid cell (horizontally or vertically) are not combined, but treated separately in the validation to avoid problems with potential artifacts resulting from merging sensors with different absolute values and gaps in their records. For the location of the in situ sites selected in this study, we refer to Figs. 4 and 7 (see further).

## 4   Results and discussion

### 4.1   Validation of soil moisture

#### 4.1.1   Inter-product differences in soil moisture accuracy

Table 3 summarizes the average correlation ($R$) and unbiased root mean square difference (ubRMSD) for the v3 soil moisture data sets (see also Table 2) against the in situ measurements. Validation statistics are calculated using all available in situ

measurements within the spatio-temporal domain of each of the data sets, as well as using a common set of soil moisture sites and an overlapping period for the three data sets (i.e. 2011–2014). Statistics for the same data sets obtained using GLEAM v2 (same input data, except for the land cover fractions) are shown between brackets for comparison. Note that the first year of each of the data sets is not taken into account for this validation exercise to avoid the effects of the model initialization on the validation statistics.

As indicated by the statistics in Table 3, all data sets compare reasonably well against the in situ measured soil moisture, with correlations for the first model layer ($w^{(1)}$) of 0.64, 0.61 and 0.63 for the v3.0a, v3.0b and v3.0c data sets, respectively. For the second model layer ($w^{(2)}$), correlations are lower (ranging from 0.49 to 0.53 for the satellite-based and long-term data set, respectively), which can be expected given the lower representativeness of a single measurement over the thicker model layer (10–100 cm). The ubRMSD yields mean values of approximately 0.06 m$^3$ m$^{-3}$ and 0.05 m$^3$ m$^{-3}$ for the first and second

model layer, respectively.

The validation statistics shown in Table 3 point to a higher quality of the v3.0a soil moisture data set as compared to the fully satellite-based data sets (i.e. v3.0b and v3.0c). This is also confirmed by the statistics obtained for the common validation set, which makes an objective comparison of the quality possible. It is shown that for both model layers and in terms of correlations, the v3.0a soil moisture is superior to the fully satellite-based data sets. Permutations of the precipitation forcing amongst the

different data sets indicate that the higher quality of the soil moisture in v3.0a is primarily due to the precipitation forcing used in each data set (not shown), and suggests an overall high accuracy of the MSWEP forcing as indicated by Beck et al. (2016). We note, however, that more than 75 % of the soil moisture probes are located in the CONUS (Continental United States),



where gauge-based precipitation data sets are known to over-perform satellite-based products (Beck et al., 2016). Recall that the TMPA 3B42v7 product is only bias-corrected using gauges, while MSWEP incorporates gauge measurements. Finally, the difference in quality between both satellite-based data sets (v3.0b and v3.0c) is relatively small, with slightly better statistics for the v3.0c data set, which integrates SMOS data.

For comparison, Table 3 also reports the validation statistics for the same data sets obtained using GLEAM v2. Both the ubRMSD and the correlations suggest that the v3 soil moisture data sets have a higher quality. This is more pronounced for the second model layer, mainly as a result of both the improved drainage formulation and the optimized data assimilation algorithm. In contrast to the assimilation of soil moisture observations in all model layers in the previous version (Miralles et al., 2014b; Martens et al., 2016), only the first model layer is updated in the new version. The latter choice is motivated by

the slower dynamics of the thick (90 cm) second model layer, which are strongly perturbed when soil moisture observations are assimilated into this layer. The impact of the soil moisture update is thus propagated towards deeper layers by drainage processes only, which ensures a smooth and natural transition of water through the profile.

Figure 4 shows maps of the difference in $R$ against in situ measured surface soil moisture for the v3 and v2 data sets. Since most in situ sites are located in the CONUS domain, also a detailed view of the results over this area is presented. As illustrated

in these maps, the quality of the soil moisture data sets improves in most regions and for the majority of sites (blue colour). It could be argued that in the Great Plains, the performance of GLEAM v3 is lower, yet only a limited number of sites are available in this area.

To evaluate the skill of GLEAM v3 to capture the effect of specific precipitation events on the estimated soil moisture – without the influence of the seasonal cycle – correlations between the anomaly time series of GLEAM soil moisture and the

anomaly time series of in situ measured soil moisture are also calculated ($R_{an}$ in Table 4). Note that to calculate a robust climatology, only in situ sites with at least 5 years of data were used, resulting in a lower subset of stations (see Table 4). As expected, correlations decay after the removal of the seasonal cycle, but remain in the range of 0.48–0.53 for the first layer and 0.41–0.47 for the second layer. In addition, results shown in Table 4 confirm the higher accuracy of the v3.0a soil moisture as compared to the fully satellite-based v3.0b data set, and the higher performance of GLEAM v3 over the previous version.

### 4.1.2   Impact of the data assimilation system

The left-hand side panel in Fig. 5 shows the differences in the correlations against the in situ measurements when data assimilation of satellite soil moisture is included in GLEAM v3 versus when it is not (i.e. model open loop). As more than 75% of soil moisture validation sites are located in the CONUS, only the results for this region are shown here. For the v3.0a data set, the assimilation of the CCI soil moisture has a rather neutral to negative impact on the modelled soil moisture states of

the first model layer. Generally, correlations are decreasing (red colour) after assimilation in very dry (e.g. West Coast of the CONUS) and forested regions (e.g. East Coast of the CONUS). In regions of limited topography and dominated by sparse vegetation (e.g. Great Plains), the quality of the modelled soil moisture is slightly improving (blue colour). For the v3.0b and v3.0c datasets, the assimilation of satellite-derived soil moisture (ESA CCI v2.3 in v3.0b and SMOS L3 in v3.0c) has – in general – a more pronounced and positive impact (blue colour) on the modelled soil moisture, especially in sparsely-vegetated





areas. The latter can be expected given the higher quality of microwave soil moisture retrievals in regions with low vegetation cover (Dorigo et al., 2015).

The negative impact of assimilating satellite observations of surface soil moisture in the v3.0a data set is partly explained by the relatively low quality of the satellite soil moisture data set (ESA CCI SM v2.3) compared to the GLEAM open-loop

soil moisture, which is of high quality in these regions largely due to the accuracy of the MSWEP precipitation forcing in the CONUS domain. This is illustrated in the central panel in Fig. 5, where difference maps between the correlations against in situ measurements of the satellite soil moisture observations and the three open-loop data sets are shown. The higher quality of the model open-loop soil moisture in terms of correlations is highlighted for some regions, including the East and West coasts of the CONUS (red colour); similar patterns are obtained for the ubRMSD (not shown). In those regions, the assimilation of satellite

soil moisture may decrease the model performance (see correspondence to left panel in Fig. 5). For the v3.0b and v3.0c data sets, the difference in correlations between satellite soil moisture and open loop is less pronounced and becomes even positive in regions with low vegetation (see central panels in Fig. 5), pointing to the higher quality of the satellite-based soil moisture observations as compared to the model estimates in those areas (e.g. Central part of the CONUS). These maps point again to the above-mentioned lower quality of the v3.0b and v3.0c precipitation forcing in those regions (TMPA 3B42v7 as opposed to

MSWEP). The subtle differences between the validation results for the v3.0b and v3.0c data sets relate to the different quality of the CCI and SMOS soil moisture observations, respectively. However, caution should be taken given the different study period and number of in situ stations used in these figures. Analogous results for anomaly time series are summarised in Fig. 6 and point to the same conclusions as drawn from Fig. 5.

Finally, it may be argued that differences in quality between the satellite-derived and modelled soil moisture should reflect

in the TCA-based quality factor ($\gamma$) used in the data assimilation algorithm (see Sect. 2.2.2). As outlined in Sect. 2.2.2, the quality factor used in the Newtonian Nudging algorithm is estimated on a yearly basis by applying a TCA on the soil moisture anomalies of three independent data sets. Based on Eq. 4 it can be seen that values of $\gamma$ below (above) 0.5 point to a lower (higher) model error relative to the observation error. The multi-year average quality factor for each of the three data sets is shown in the right-hand side panel in Fig. 5. Spatial patterns in these maps agree well with the ones observed in the central maps,

reflecting the ability of the TCA to capture the relative errors of modelled and observed surface soil moisture. Nonetheless, despite the overall low quality factors for the v3.0a data set (i.e. $\gamma$ rarely exceeds 0.3) – which reflects the higher error of the observed soil moisture relative to the model open loop – a decrease in quality is often observed when this soil moisture data set is assimilated into GLEAM v3.0a (see discussion above). As expected, the quality factors for the v3.0b and v3.0c data sets are higher and exceed 0.5 in some low-vegetated regions, indicating again the higher quality of the satellite-based soil moisture

observations as compared to the model open loop in these areas (see discussion above).

Therefore, the inability of the assimilation algorithm to consistently improve the v3.0a soil moisture in regions such as the East and West coasts of the CONUS is mainly explained by the substantially higher quality of the model open loop. Nevertheless, our simple Newtonian Nudging data assimilation system will still correct for random forcing errors and other effects such as irrigation, that are not explicitly modelled in GLEAM. Moreover, the Newtonian Nudging scheme minimizes





the computational costs and, as shown by Miralles et al. (2014b) and Lievens et al. (2016), the use of more complex Kalman filters does not significantly improve the model performance.

## 4.2 Validation of evaporation

### 4.2.1 Inter-product differences in evaporation accuracy

Table 5 lists the validation statistics for the different evaporation data sets. In contrast to the results of the soil moisture validation exercise (see Table 3), differences between the three data sets are less pronounced when diagnosing the quality of their evaporation estimates. For the overlapping period 2011–2014 and the common sample of sites, an average correlation of 0.78, and a similar ubRMSD of approximately 0.73 mm day$^{-1}$ is obtained for all three data sets.

Analogous statistical inferences for the validation of GLEAM v2 are shown between brackets and differ only slightly from
the ones calculated for the data sets obtained using the new model version. Figure 7 shows maps of the differences in correlation against the in situ measurements for the v3 and v2 data sets. It can be seen from the maps for the v3.0a and v3.0b data sets that GLEAM v2 performs generally better along the Eastern part of the CONUS, which may be attributed to a lower performance of the stress functions implemented in the new v3, since the v3 soil moisture is consistently improved in this region (see Fig. 4). However, given the low number of in situ sites, no clear conclusions on geographical patterns can be drawn until the foreseen
extension of the FLUXNET 2015 synthesis database (http://fluxnet.fluxdata.org/). Over Continental Australia, GLEAM v3 performs generally better, except for the v3.0c data set, where for some sites a deterioration of the results is shown. However, as the validation database for the latter data set contains a significantly lower number of measurements, due to the shorter time period, it may be less representative of the overall quality of the data set. Also, note that the results obtained for the three data sets do not necessarily agree at each in situ site, suggesting an impact of the validation period. Correlations for the anomaly
time series are listed in Table 6 and confirm the above conclusions.

As an example, Fig. 8 shows time series of GLEAM and in situ measured evaporation for two validation sites, i.e. US-Ne3 (Central part of the CONUS, see Table A1) at the left-hand side and AU-ASM (Central Australia, see Table A1) at the right-hand side. While for the first site the performance of GLEAM v3 tends to be lower, statistics are improving for the second site with respect to the previous version of the model. However, time series show an overall good correspondence between
model and in situ measurements. For US-Ne3, correlations drop from 0.82 (v2.0a) and 0.83 (v2.0b) to 0.78 (v3.0a) and 0.77 (v3.0b); on the other hand, for the SMOS-based data sets (v2.0c and v3.0c), correlations increase from 0.73 to 0.76. Analogous differences are obtained in terms of ubRMSD. Despite the apparent decrease in quality for the v3 data sets, the time series shown in Fig. 8 illustrate that the estimates of evaporation are realistic and have no systematic errors. For AU-ASM, correlations consistently improve for all three data sets from 0.84 (v2.0a), 0.84 (v2.0b) and 0.78 (v2.0c) to 0.88 (v3.0a), 0.88 (v3.0b) and
0.84 (v3.0c), and similar improvements are obtained for the ubRMSD. Time series at the right-hand side of Fig. 8 show that the better results are mainly explained by the improved estimates of the evaporative flux during the dry season. For these periods, GLEAM v3 estimates lower volumes of evaporation, resulting in a closer agreement with the in situ measurements. This is mainly related to the new drainage formulation, which allows a faster dry-out during precipitation-free periods, leading



to an increase in the evaporative stress. Additionally, the new drainage algorithm also yields less extreme evaporation peaks after precipitation events, since the faster drainage implies that the soil profile requires stronger precipitation events to saturate. Results for the Australian site indicate that these evaporation patterns are realistic under conditions of water stress, yet caution may be taken when extrapolating these findings to other climatic and ecological regimes.

### 4.2.2 Global magnitude and variability of terrestrial evaporation

The top row in Fig. 9 presents the mean annual evaporation from the v3.0a data set (left) and a difference map with the v2.0a data set (right). Analogous results are obtained for the v3.0b and v3.0c data sets, but are not shown here. As expected, the general climatic patterns of evaporation are captured well by both data sets. Compared to the numbers and patterns reported in Miralles et al. (2016a), based on five models and different forcing data, the maps shown in Fig. 9 appear realistic. Differences in the annual totals between v3.0a and v2.0a amount to 100 mm y$^{-1}$ in several regions, with overall less evaporation in areas covered by short vegetation and more evaporation in desert-like and tropical regions for the new version. The total continental evaporation (excluding inland water bodies) amounts to $66 \cdot 10^3$ km$^3$ (v3.0a) versus the $68 \cdot 10^3$ km$^3$ from the previous version (v2.0a); these numbers agree well with previously reported values from a range of independent sources (see Miralles et al. (2016a) and references therein).

The remaining maps in Fig. 9 show the partitioning of GLEAM evaporation in its different components; i.e. forest interception loss, transpiration and bare-soil evaporation. Note that for illustrative purposes only, the estimated sublimation is added to the bare-soil flux and the evaporation from inland waters (open-water evaporation) is not considered here. Averaged over the entire land surface, approximately 74% of the total flux of water from land into the atmosphere is coming from transpiration, 15% comes from bare-soil evaporation and about 11% is the result of interception loss; for the v2.0a data set, 80%, 8% and 12% are obtained, respectively. These discrepancies are also evidenced in the difference maps shown in the right-hand side panel in Fig. 9. It can be seen that almost across the entire globe the bare-soil evaporation is higher in the v3.0a data set; only for some drier regions such as the Namibian desert, central Australia and parts of Chile, the bare-soil evaporation is decreased. In contrast, transpiration typically increases in these areas. As shown, the total flux of interception loss is generally lower in the new version, except for some parts of Amazonia, Eastern China and the CONUS where a clear increase may be observed. All these differences are the result of the modified stress functions, but – more importantly – of the new (high-resolution) land cover fractions used in GLEAM v3 which report an overall larger portion of bare soils over the continents (see Sect. 3). The higher contribution of bare-soil evaporation and the lower volumes of transpiration result in closer agreement with the partitioning obtained from other data sets, especially in semi-arid regions like the Sahel (Wang et al., 2014; Schlesinger and Jasechko, 2014; Miralles et al., 2016a; Good et al., 2015). However, Miralles et al. (2016a) recently raised awareness about the use of satellite-based evaporation algorithms to assess the contribution from different evaporation components, and suggested to avoid the use of any single model in isolation.



## 5   Conclusions

The available stack of satellite-derived geophysical variables related to the process of evaporation – such as soil moisture, air temperature and net radiation – is continuously growing and improving. As a result, models aiming at the accurate estimation of terrestrial evaporation from satellite observations need to be updated to optimally incorporate these new data. Concurrently, as our knowledge of the relevant physical processes advances based on new experimental evidence, these simple retrieval models can increase their realism. With the overarching goal of improving our understanding of continental evaporation, a next version of the Global Land Evaporation Amsterdam Model (GLEAM v3) – a set of algorithms dedicated to the estimation of global terrestrial evaporation from satellite data – is presented in this paper.

Three major modifications are included in this new version: (1) a revised representation of the evaporative stress, (2) an optimized water-balance module, and (3) a new soil moisture data assimilation strategy. Using GLEAM v3 and different forcing data sets, three novel data sets of root-zone soil moisture and terrestrial evaporation are presented. The first data set (v3.0a) spans the 35-year period 1980–2014, has a global coverage, and is produced using satellite-observed soil moisture, vegetation optical depth and snow water equivalents, reanalysis air temperature and radiation, and a multi-source precipitation product. The remaining two data sets (v3.0b and v3.0c) are based on satellite forcing exclusively, with the only difference the use of SMOS-based VOD and soil moisture in the v3.0c data set, as opposed to the use of the corresponding CCI forcing in the v3.0b data set. Both satellite-based data sets are quasi-global (50°N–50°S) and span a significantly shorter period (2003–2015 for v3.0b and 2011–2015 for v3.0c).

Results based on the validation of these three data sets against an extensive set of in situ measured evaporation and soil moisture point to a slightly higher quality of the v3.0a soil moisture data set as compared to the other two data sets, while the quality of the modelled evaporation is rather similar across all three. The higher accuracy of the v3.0a soil moisture is explained by the high quality of the MSWEP precipitation forcing over the regions where soil moisture probes are located, compared to the satellite-based forcing in the v3.0b and v3.0c data sets. Results, however, might be biased given that the majority (i.e. more than 75%) of the in situ soil moisture sites are located in the CONUS, where gauge-based precipitation products are know to perform better (Beck et al., 2016).

The quality of the new v3 data sets is also compared to analogous data sets obtained using GLEAM v2. It is shown that for the soil moisture, the modifications in GLEAM result in a consistent improvement of soil moisture across the vertical profile. These improvements mainly relate to the optimized drainage algorithm and the new data assimilation system, which allow a more realistic representation of the downward flux of water through the soil profile. On the other hand, the increased quality of the evaporation data is not clear from the in situ validation, likely hampered by the low availability of validation sites. It is illustrated that, on average, the performance of GLEAM v3 is comparable to that of the former version.

On top of the modifications in the algorithms, the static data set describing the land cover fractions per pixel is also updated. The effect of this change is investigated through the analysis of the partitioning of terrestrial evaporation into its different components, which shows an increase in bare-soil evaporation almost in every continental region, while interception loss



generally decreases. An increase in transpiration can be observed for some dry regions such as the Namibian desert and Central Australia.

Based on the results in this study, it can be concluded that the modifications in GLEAM have led to a more realistic representation of physical processes and an overall increased quality of the data sets. Following the advances in satellite technology and

5    the increased availability of these data, GLEAM will be further optimized in coming years. Future activities may concentrate on the incorporation of novel data sets (e.g. high-resolution microwave VOD and surface soil moisture, solar-induced fluorescence, land-surface temperature), the application of GLEAM to higher resolutions and in near-real time, and the improved partitioning of evaporation into its different components. Meanwhile, all the novel data sets of observation-based terrestrial evaporation and root-zone soil moisture presented in this study are now available for studies of hydrological cycle dynamics

10   and climate model benchmarking using www.GLEAM.eu as gateway.

## 6   Code and data availability

The model code of GLEAM v3 is available upon request from the corresponding author. Datasets described in this paper can be freely accessed using www.GLEAM.eu as gateway.



## Appendix A: *In situ* eddy-covariance sites

**Table A1.** List of the FLUXNET sites used in this study together with their IGBP land cover and official reference (or principal investigator (PI)).

| Station | Land Cover | Reference/PI | Station | Land Cover | Reference/PI |
|---|---|---|---|---|---|
| AU-Stp | GRA | Beringer (2013c) | DE-Kli | CRO | Christian Bernhofer |
| AR-Vir | ENF | Posse et al. (2016) | DE-Lkb | ENF | Rainer Steinbrecher |
| AU-ASM | ENF | Cleverly (2011) | DE-Obe | ENF | Christian Bernhofer |
| AU-Cpr | SAV | Calperum Tech (2013) | DE-Spw | WET | Christian Bernhofer |
| AU-DaP | GRA | Beringer (2013a) | DE-Tha | ENF | Christian Bernhofer |
| AU-DaS | SAV | Beringer (2013d) | FI-Hyy | ENF | Timo Vesala |
| AU-Dry | SAV | Beringer (2013b) | FI-Jok | CRO | Tuomas Laurila |
| AU-Emr | GRA | Schroder (2014) | FR-Gri | CRO | Pierre Cellier |
| AU-GWW | SAV | Macfarlane (2013) | IT-La2 | ENF | Alessandro Cescatti |
| AU-RDF | WSA | Beringer (2014b) | IT-PT1 | DBF | Günther Seufert |
| AU-Rig | GRA | Beringer (2014a) | IT-Ren | ENF | Stefano Minerbi |
| AU-Tum | EBF | van Gorsel (2013) | RU-Fyo | ENF | Milyukova et al. (2002) |
| AU-Whr | EBF | Beringer (2013f) | RU-Skp | DNF | Trofim Maximov |
| AU-Ync | UNK | Beringer (2013e) | SD-Dem | SAV | Ardö et al. (2008) |
| BE-Bra | MF | Ivan Janssens | US-ARM | CRO | Fischer et al. (2007) |
| BR-Sa3 | EBF | Steininger (2004) | US-Blo | ENF | Goldstein et al. (2000) |
| CA-Gro | MF | McCaughey et al. (2006) | US-Goo | GRA | Tilden Meyers |
| CA-NS7 | OSH | Bond-Lamberty et al. (2004) | US-Ha1 | DBF | Goulden et al. (1996) |
| CH-Fru | GRA | Zeeman et al. (2010) | US-MMS | DBF | Schmid et al. (2000) |
| CH-Oe1 | GRA | Christof Ammann | US-Me6 | ENF | Ruehr et al. (2012) |
| CN-Cha | MF | Shijie Han | US-Ne1 | CRO | Simbahan et al. (2006) |
| CN-Cng | GRA | Gang Dong | US-Ne2 | CRO | Amos et al. (2005) |
| CN-Dan | GRA | Shi Peili | US-Ne3 | CRO | Verma et al. (2005) |
| CN-Din | EBF | Guoyi Zhou | US-Oho | DBF | Noormets et al. (2008) |
| CN-Du2 | GRA | Shiping Chen | US-SRM | WSA | Scott et al. (2009) |
| CN-Ha2 | WET | Yingnian Li | US-Ton | WSA | Chen et al. (2007) |
| CN-HaM | GRA | Kato et al. (2006) | US-Tw3 | CRO | Dennis Baldocchi |
| CN-Qia | ENF | Huimin Wang | US-Var | GRA | Ma et al. (2007) |
| CZ-BK1 | ENF | Marian Pavelka | US-WCr | DBF | Cook et al. (2004) |
| CZ-BK2 | GRA | Marian Pavelka | US-Whs | OSH | Scott (2010) |
| DE-Gri | GRA | Christian Bernhofer | US-Wkg | GRA | Scott et al. (2010) |
| DE-Hai | DBF | Knohl et al. (2003) | ZA-Kru | SAV | Bob Scholes |



*Author contributions.* All authors have been involved in interpreting the results, discussing the findings, and editing the manuscript. B.M., H.L. and D.G.M. implemented GLEAM v3. B.M., R.v.d.S., H.E.B and W.A.D. processed the forcing and validation data. B.M. and D.G.M. designed the lay-out of the paper and wrote the draft of the manuscript.

*Acknowledgements.* This work has been funded by ESA's Support To Science Element through the SMOS+ET II project (IPL-POE-2015-
5  723-LG-cb-LE). D.G.M. acknowledges the financial support from The Netherlands Organization for Scientific Research through grant 863.14.004. H.L. is a postdoctoral research fellow of the Research Foundation Flanders (FWO). W.A.D. is supported by the personal grant "TU Wien Wissenschaftspreis 2015" and ESA's Climate Change Initiative for Soil Moisture (Contract No. 4000112226/14/I-NB). The authors would like to thank the principal investigators of the International Soil Moisture Network (ISMN). This work used eddy covariance data acquired and shared by the FLUXNET community, including these networks: AmeriFlux, AfriFlux, AsiaFlux, CarboAfrica, CarboEuropeIP,
10  CarboItaly, CarboMont, ChinaFlux, Fluxnet-Canada, GreenGrass, ICOS, KoFlux, LBA, NECC, OzFlux-TERN, TCOS-Siberia, and USCCC. The FLUXNET eddy covariance data processing and harmonization was carried out by the ICOS Ecosystem Thematic Center, AmeriFlux Management Project and Fluxdata project of FLUXNET, with the support of CDIAC, and the OzFlux, ChinaFlux and AsiaFlux offices. The authors also acknowledge Instituto Nacional Technologica Agropecuaria for making the eddy-covariance data of AR-Vir publically available.





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

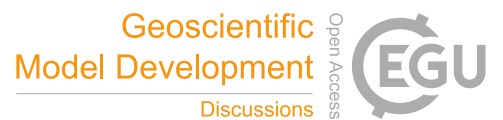

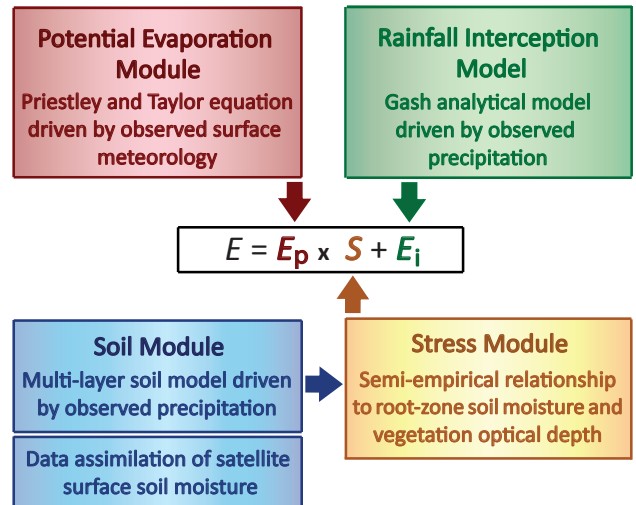

**Figure 1.** Schematic of the four modules of GLEAM.

**Table 1.** List of the selected forcing data sets together with their references, the original spatial resolution and period of availability.

| Variable | Data set | Type | Resolution | Period | References |
|---|---|---|---|---|---|
| Radiation Components | CERES L3SYN1DEG | Satellite | 1° | 2001–2015 | Wielicki (1996) |
| | ERA-Interim | Reanalysis | 0.75° | 1979–2015 | Dee et al. (2011) |
| Precipitation | TMPA 3B42v7 | Merge | 0.25° | 1998–2015 | Huffman et al. (2007) |
| | MSWEP v1.0 | Merge | 0.25° | 1979–2014 | Beck et al. (2016) |
| Air Temperature | AIRS L3RetStdv6.0 | Satellite | 1° | 2003–2015 | Aumann et al. (2003) |
| | ERA-Interim | Reanalysis | 0.75° | 1979–2015 | Dee et al. (2011) |
| Snow Water Equivalent | GLOBSNOW L3av2 + NSIDC v0.1 | Satellite | 0.25° | 1980–2015 | Luojus et al. (2013) |
| | | | | | Armstrong et al. (2005) |
| VOD | SMOS-LPRM | Satellite | 25 km | 2011–2015 | van der Schalie et al. (2015, 2016) |
| | CCI-LPRM | Satellite | 0.25° | 1980–2012 | Liu et al. (2011, 2013) |
| Soil Moisture | SMOS L3 | Satellite | 25 km | 2010–2015 | Jacquette et al. (2010) |
| | ESA CCI SM v2.3 | Satellite | 0.25° | 1978–2015 | Liu et al. (2012); Wagner et al. (2012) |
| | GLDAS Noah | Reanalysis | 1° | 1980–2015 | Rodell et al. (2004) |
| Cover Fractions | MOD44B | Satellite | 250 m | static | Hansen et al. (2005) |
| Soil Properties | IGBP-DIS | Survey | 0.25° | static | Global Soil Data Task Group (2000) |
| Lightning Frequency | LIS/OTD | Satellite | 5 km | static | Mach et al. (2007) |



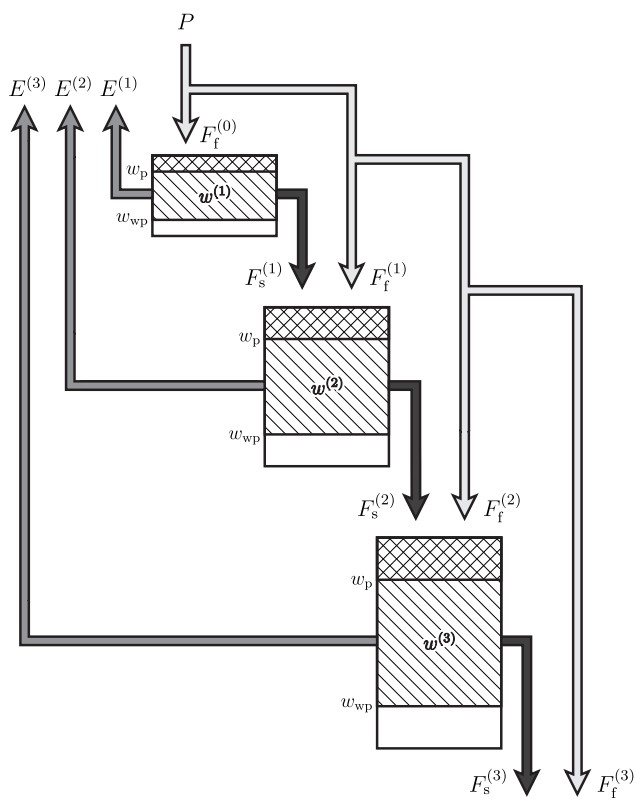

**Figure 2.** Schematic of the water-balance module implemented in GLEAM v3 for the fraction of tall vegetation. $w^{(l)}$ (m$^3$ m$^{-3}$) is the volumetric soil moisture content of layer $l$, $F_s^{(l)}$ (mm/day) is the slow draining volume of water, $F_f^{(l)}$ (mm/day) is the fast draining volume of water, $E^{(l)}$ (mm/day) is the evaporative flux, $P$ (mm/day) is the net precipitation, $w_{wp}$ (m$^3$ m$^{-3}$) is the wilting point and $w_p$ (m$^3$ m$^{-3}$) is the porosity.

**Table 2.** Overview of the forcing data used to produce the three GLEAM data sets of terrestrial evaporation and root-zone soil moisture.

| Data set | Coverage | Period | Radiation | Precipitation | Air Temperature | VOD | Soil Moisture |
|----------|----------|--------|-----------|---------------|-----------------|-----|---------------|
| v3.0a | Global | 1980–2014 | ERA-Interim | MSWEP | ERA-Interim | CCI/SMOS-LPRM | ESA CCI SM |
| v3.0b | 50°N–50°S | 2003–2015 | AIRS | TMPA | AIRS | CCI/SMOS-LPRM | ESA CCI SM |
| v3.0c | 50°N–50°S | 2011–2015 | AIRS | TMPA | AIRS | SMOS-LPRM | SMOS L3 |





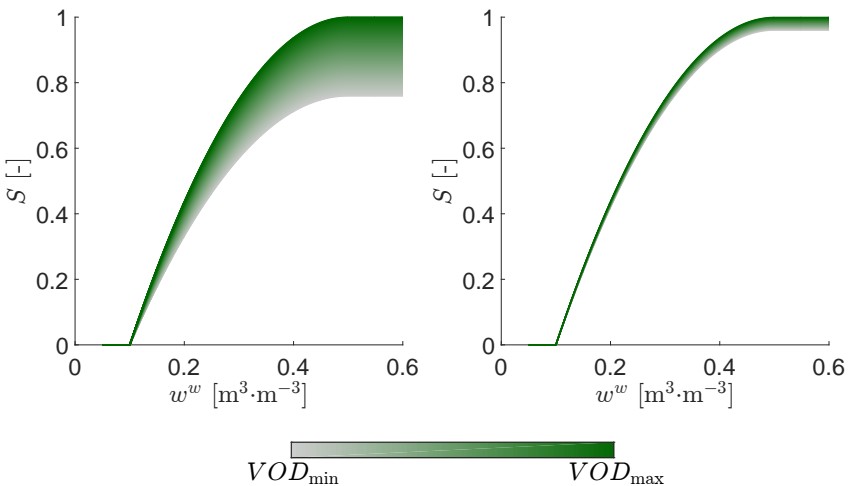

**Figure 3.** Illustration of the stress function implemented in GLEAM v3 for the fractions of tall and short vegetation (colours relate to the VOD). Left-hand side: high range in VOD ($\text{VOD}_{\text{max}} = 0.47$, $\text{VOD}_{\text{min}} = 0.27$); right-hand side: low range in VOD ($\text{VOD}_{\text{max}} = 0.40$, $\text{VOD}_{\text{min}} = 0.38$). For illustrative purposes only, the wilting point and the critical soil moisture for both figures were set to 0.1 and 0.5 m$^3$ m$^{-3}$, respectively.

**Table 3.** Average validation statistics for the different soil moisture data sets (v3.0a, v3.0b and v3.0c) and for the first two model layers ($w^{(1)}$ and $w^{(2)}$) against in situ measurements: ubRMSD is the unbiased root mean square difference, $R$ is the correlation and $N$ is the number of sites included in the sample. The first part of the table reports the averaged statistics over all available sites and the entire study period (see also Table 2), while the second part shows the same statistics for a common sample of sites, and an overlapping study period (2011–2014) for the three data sets. The same statistics for the data sets produced using GLEAM v2 are reported between brackets.

| Data set | Layer | Complete record | | | Overlap period | | |
|---|---|---|---|---|---|---|---|
| | | $N$ | ubRMSD | $R$ | $N$ | ubRMSD | $R$ |
| | | [–] | [m$^3$ m$^{-3}$] | [–] | [–] | [m$^3$ m$^{-3}$] | [–] |
| v3.0a | $w^{(1)}$ | 1119 | 0.059 (0.060) | 0.64 (0.61) | 782 | 0.059 (0.062) | 0.65 (0.61) |
| | $w^{(2)}$ | 1219 | 0.048 (0.051) | 0.53 (0.47) | 748 | 0.048 (0.054) | 0.51 (0.43) |
| v3.0b | $w^{(1)}$ | 1038 | 0.061 (0.062) | 0.61 (0.58) | 782 | 0.061 (0.063) | 0.61 (0.58) |
| | $w^{(2)}$ | 1127 | 0.049 (0.052) | 0.49 (0.42) | 748 | 0.049 (0.054) | 0.48 (0.41) |
| v3.0c | $w^{(1)}$ | 784 | 0.059 (0.063) | 0.63 (0.58) | 782 | 0.059 (0.063) | 0.63 (0.58) |
| | $w^{(2)}$ | 754 | 0.048 (0.052) | 0.49 (0.42) | 748 | 0.048 (0.053) | 0.49 (0.42) |





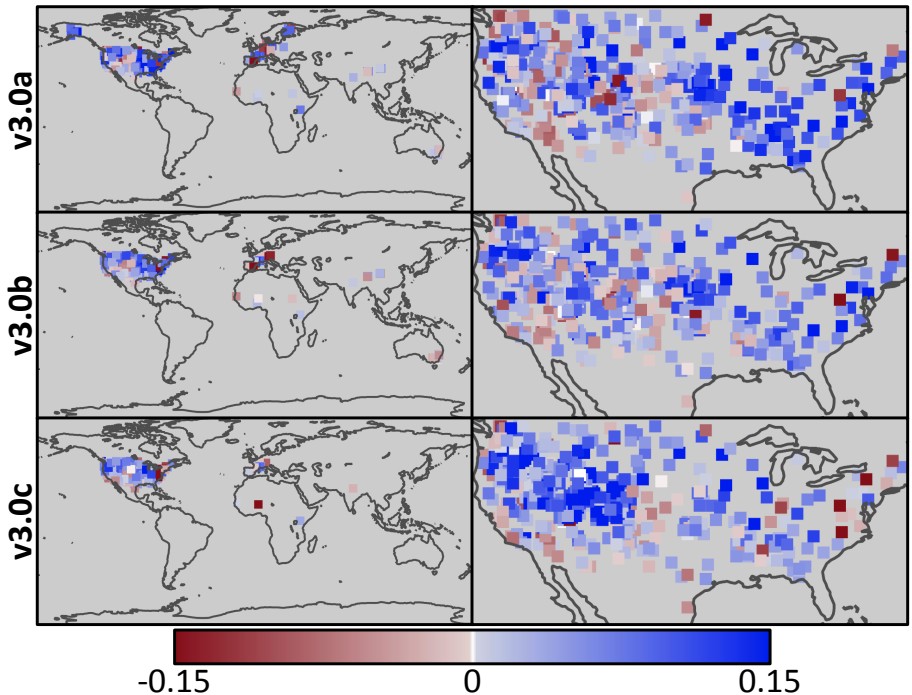

**Figure 4.** Difference in quality between the v3 and v2 data sets of surface soil moisture ($R(\mathrm{GLEAM\,v3, in\,situ}) - R(\mathrm{GLEAM\,v2, in\,situ})$). Colours relate to the difference in correlations against in situ measurements for the v3 and v2 surface soil moisture data sets. Statistics are calculated based on all available sites reporting measurements falling within the spatio-temporal domain of the different data sets. Maps at the right show a detailed overview of the results for the CONUS.

**Table 4.** Average anomaly correlations for different soil moisture data sets (v3.0a and v3.0b) and for the first two model layers ($w^{(1)}$ and $w^{(2)}$) against in situ measurements: $R_{\mathrm{an}}$ is the anomaly correlation and $N$ is the number of sites included in the sample. The first part of the table reports the averaged statistics over all available sites and the entire study period (see also Table 2), while the second part shows the same statistics for a common sample of sites, and an overlapping study period (2004–2014) for the two data sets. The same statistics for the data sets produced using GLEAM v2 are reported between brackets.

| Data set | Layer | Complete record | | Overlap period | |
|---|---|---|---|---|---|
| | | $N$ | $R_{\mathrm{an}}$ | $N$ | $R_{\mathrm{an}}$ |
| | | [–] | [–] | [–] | [–] |
| v3.0a | $w^{(1)}$ | 515 | 0.53 (0.50) | 455 | 0.54 (0.52) |
| | $w^{(2)}$ | 714 | 0.47 (0.42) | 622 | 0.46 (0.43) |
| v3.0b | $w^{(1)}$ | 455 | 0.48 (0.47) | 455 | 0.48 (0.47) |
| | $w^{(2)}$ | 623 | 0.41 (0.39) | 622 | 0.41 (0.39) |





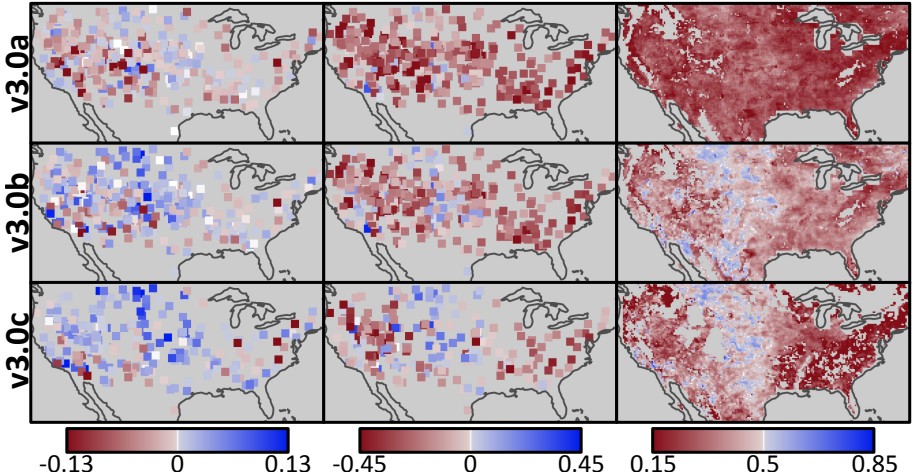

**Figure 5.** Impact of the data assimilation system in GLEAM v3 on the surface soil moisture for the CONUS. Left-hand side figures show the difference in correlations against in situ measurements for the GLEAM v3 surface soil moisture data sets with and without (open loop) the assimilation of satellite-derived soil moisture ($R(\mathrm{DA}, \mathrm{in\ situ}) - R(\mathrm{OL}, \mathrm{in\ situ})$). Maps in the central panel show the difference in correlations against in situ measurements for the satellite-derived soil moisture data sets and the v3 soil moisture data sets without data assimilation ($R(\mathrm{SAT}, \mathrm{in\ situ}) - R(\mathrm{OL}, \mathrm{in\ situ})$). Maps at the right show the quality factor $\gamma$ calculated in the data assimilation system. The latter balances both the model and observation errors, with values above (below) 0.5 indicating a lower (higher) error in the observations relative to GLEAM.

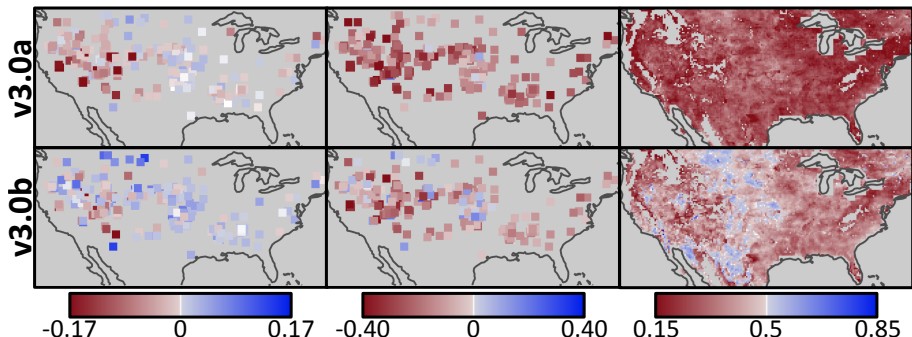

**Figure 6.** Impact of the data assimilation system in GLEAM v3 on the surface soil moisture for the CONUS. Left-hand side figures show the difference in anomaly correlations against in situ measurements for the GLEAM v3 surface soil moisture data sets with and without (open loop) the assimilation of satellite-derived soil moisture ($R_{\mathrm{an}}(\mathrm{DA}, \mathrm{in\ situ}) - R_{\mathrm{an}}(\mathrm{OL}, \mathrm{in\ situ})$). Maps in the central panel show the difference in anomaly correlations against in situ measurements for the satellite-derived soil moisture data sets and the GLEAM v3 soil moisture data sets without data assimilation ($R_{\mathrm{an}}(\mathrm{SAT}, \mathrm{in\ situ}) - R_{\mathrm{an}}(\mathrm{OL}, \mathrm{in\ situ})$). Maps at the right show the quality factor $\gamma$ calculated in the data assimilation system. The latter balances both the model and observations errors, with values above (below) 0.5 indicating a lower (higher) error in the observations relative to GLEAM.





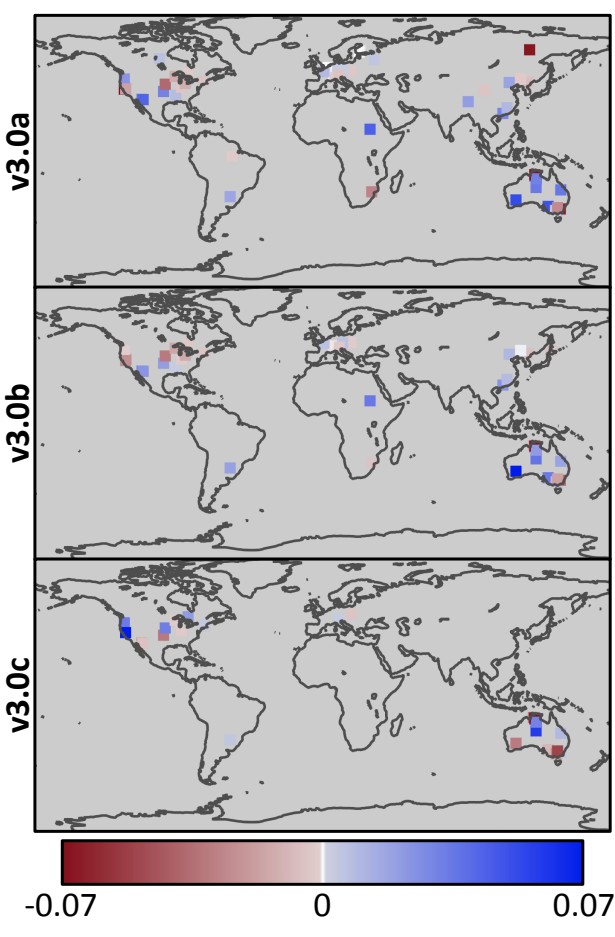

**Figure 7.** Difference in quality between the v3 and v2 data sets of terrestrial evaporation $(R(\text{GLEAM v3}, \text{in situ}) - R(\text{GLEAM v2}, \text{in situ}))$. Colours relate to the difference in correlations against in situ measurements for the v3 and v2 evaporation data sets. Statistics are calculated based on all available sites reporting measurements falling within the spatio-temporal domain of the different data sets.





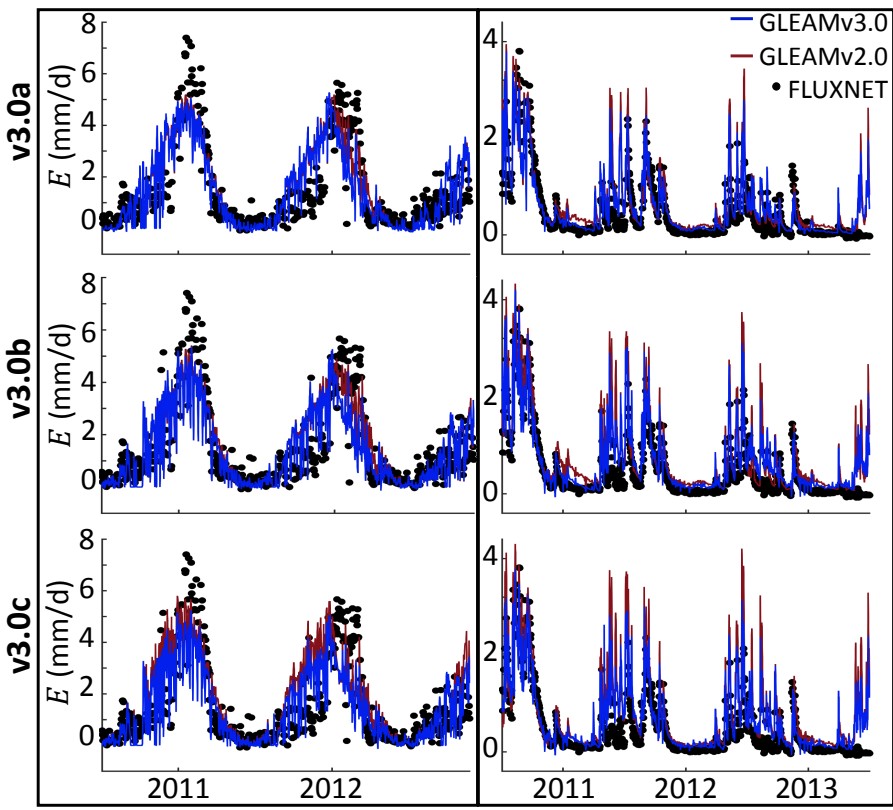

**Figure 8.** Time series of GLEAM and in situ measured evaporation for two in situ validation sites: US-Ne3 (left) and AU-ASM (right).

**Table 5.** Average validation statistics for the different evaporation data sets (v3.0a, v3.0b and v3.0c) against in situ measurements: ubRMSD is the unbiased root mean square difference, $R$ is the correlation and $N$ is the number of sites included in the sample. The first part of the table reports the averaged statistics over all available sites and the entire study period (see also Table 2), while the second part shows the same statistics for a common sample of sites, and an overlapping study period (2011–2014) for the three data sets. The same statistics for the data sets produced using GLEAM v2 are reported between brackets.

| Data set | Complete record | | | Overlap period | | |
|---|---|---|---|---|---|---|
| | $N$ | ubRMSD | $R$ | $N$ | ubRMSD | $R$ |
| | [–] | [mm day$^{-1}$] | [–] | [–] | [mm day$^{-1}$] | [–] |
| v3.0a | 65 | 0.73 (0.79) | 0.79 (0.80) | 29 | 0.72 (0.72) | 0.78 (0.78) |
| v3.0b | 46 | 0.77 (0.77) | 0.80 (0.79) | 29 | 0.74 (0.74) | 0.78 (0.78) |
| v3.0c | 30 | 0.74 (0.79) | 0.78 (0.78) | 29 | 0.73 (0.78) | 0.78 (0.78) |

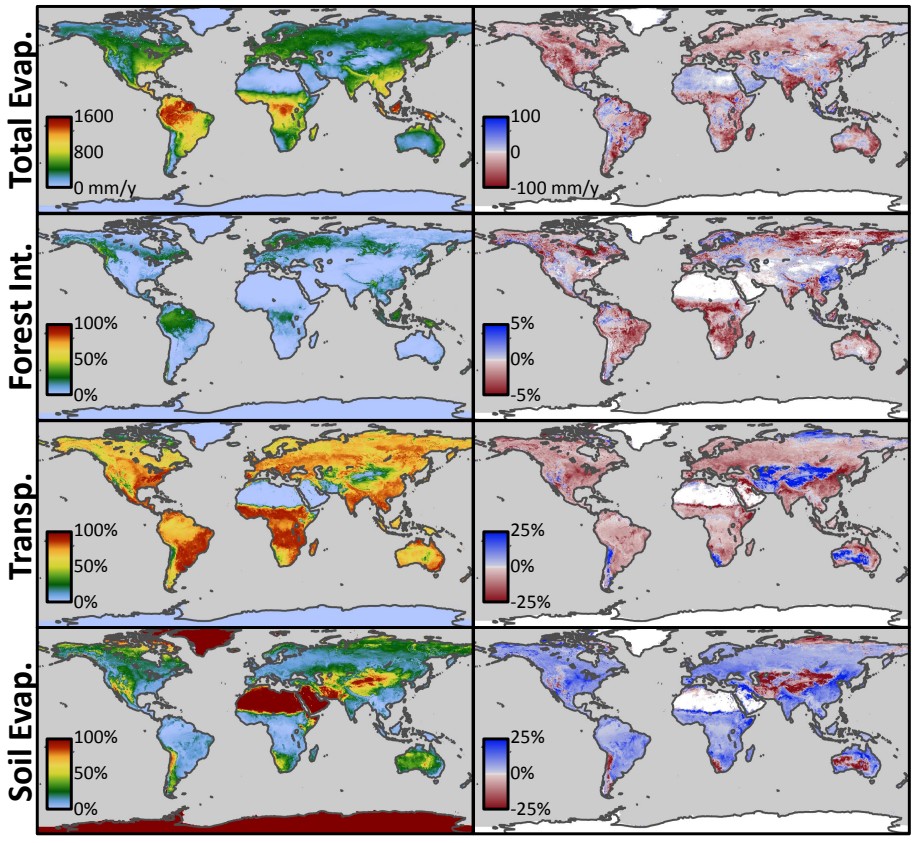

**Figure 9.** Global maps of terrestrial evaporation (top row) and the partitioning in its different components, i.e. forest interception loss (second row), transpiration (third row) and bare-soil evaporation (bottom row) for the v3.0a data set. On top, the multi-annual total flux of evaporation for the v3.0a data set (left) and the difference with the v2.0a data set (right) are shown. The other maps show the percentage of the total flux in the v3.0a data set coming from the different components (left) and the difference with the same maps for the v2.0a data set (right).



**Table 6.** Average anomaly correlations for different evaporation data sets (v3.0a and v3.0b) against in situ measurements: $R_{an}$ is the anomaly correlation and $N$ is the number of sites included in the sample. The first part of the table reports the averaged statistics over all available sites and the entire study period (see also Table 2), while the second part shows the same statistics for a common sample of sites, and an overlapping study period (2004–2014) for the two data sets. The same statistics for the data sets produced using GLEAM v2 are reported between brackets.

| Data set | Complete record | | Overlap period | |
|---|---|---|---|---|
| | $N$ | $R_{an}$ | $N$ | $R_{an}$ |
| | [–] | [–] | [–] | [–] |
| v3.0a | 34 | 0.40 (0.39) | 24 | 0.42 (0.42) |
| v3.0b | 24 | 0.48 (0.49) | 24 | 0.48 (0.49) |