# Peer review of "GLEAM v3: satellite-based land evaporation and root-zone soil moisture"

_Geoscientific Model Development, 2016_

## Short Comment (SC1) · 20 Sep 2016

I find the manuscript very well written and full of useful information both for using GLEAM algorithm and the generated datasets. This document is of high importance to document recent advances in the generation of datasets of water cycle components for climate studies.

As this paper aims at clearly and fully describing the new algorithm and validation, I have a couple of questions regarding its implementation.

- GLEAM v3 uses MOD44B product for partitioning the land cover classes. Which version and how is it used (updated regularly throughout the time period of the dataset or fixed at one specific date or else ?) In the conclusion (on p16, l.31-33),

there is mention of an update. Does it mean the land cover fraction is different from a previous GLEAM dataset ?

- Air temperature is used to force GLEAM v3. While details are given about forcing v3a (ECMWF ERA-Interim, 3h spacing), I do not find the complete information regarding the forcing of v3b and v3c. Could you indicate the source providing the dataset, and its temporal frequency?

- Thank you for providing the link to the dedicated website. I found also a visualizing tool on the H2Observe project portal (https://wci.earth2observe.eu/), it may also be mentioned.

- For the validation, if I am not mistaken, the choice of correcting here the observed fluxes using the Bowen ratio (p10, l29) differs from the strategy of validation presented in Michel et al, 2016, for evaluating WACMOS-ET datasets. Is there a reason for that ?

- Extreme outliers are screened, what is the mask applied ? For gaps in data, what is the strategy if less than 25

---

## Author Comment (AC1) · 20 Sep 2016

We would like to thank Dr. Ghilain for his interest in the manuscript and for posting some interesting comments and questions.

Attached, a detailed reply to the questions and comments raised by Dr. Ghilain can be found. The replies of the authors are highlighted in bold.

Please also note the supplement to this comment: http://www.geosci-model-dev-discuss.net/gmd-2016-162/gmd-2016-162-AC1-supplement.pdf

[Figure]

**Supplement:**

**N. Ghilain**
*nicolas.ghilain@meteo.be*

I find the manuscript very well written and full of useful information both for using GLEAM algorithm and the generated datasets. This document is of high importance to document recent advances in the generation of datasets of water cycle components for climate studies.
**We would like to thank Dr. Ghilain for his interest in the manuscript and for posting some interesting comments and questions.**

As this paper aims at clearly and fully describing the new algorithm and validation, I have a couple of questions regarding its implementation.

- GLEAM v3 uses MOD44B product for partitioning the land cover classes. Which version and how is it used (updated regularly throughout the time period of the dataset or fixed at one specific date or else?) In the conclusion (on p16, l.31-33), there is mention of an update. Does it mean the land cover fraction is different from a previous GLEAM dataset?
  **As discussed in the paper at P10_L5-8, the land cover fractions used in GLEAM are static. The product used here is the fine resolution (250 m) MOD44B v51 (the version will be added to Table 2 in the revised version of the paper). The product was upscaled to the required 0.25 degree resolution of GLEAM. While this product is available on a yearly basis, the data providers suggests to "exercise caution when considering performing inter-annual comparisons[1]". This suggest that part of the inter-annual variability observed in the land cover maps cannot be physically explained, and that the uncertainties in the temporal dynamics preclude from using the product other than as a static dataset (i.e. average of the individual yearly products). As listed at P10_L5-8, a similar coarse-scale product (0.25 degrees, provided directly by the MODIS team) was used in the previous versions of GLEAM. However, in this v3 we have chosen to upscale the higher native resolution dataset ourselves, by means of averaging the 250 m grid-cells contained in each 0.25 degree pixel.**

- Air temperature is used to force GLEAM v3. While details are given about forcing v3a (ECMWF ERA-Interim, 3h spacing), I do not find the complete information regarding the forcing of v3b and v3c. Could you indicate the source providing the dataset, and its temporal frequency?
  **We would like to point out that the details of the air temperature forcing are listed in the paper at P9_L3-5 and Tables 1 and 2. For the v3.0b and c datasets, air temperatures derived from measurements of the Atmospheric Infrared Sounder (AIRS) were processed (the AIRS L3RetStdv6.0 product is used here). These datasets are available on a daily resolution.**
* * *
[1] https://lpdaac.usgs.gov/products/modis_products_table/mod44b

- Thank you for providing the link to the dedicated website. I found also a visualizing tool on the H2Observe project portal (https://wci.earth2observe.eu/), it may also be mentioned.

  **We agree with the reviewer on the value of the visualization tool from EartH2Observe. However, since we are only responsible for the management of the records hosted at www.gleam.eu, we can only guaranty the availability of the latest product versions via this platform.**

- For the validation, if I am not mistaken, the choice of correcting here the observed fluxes using the Bowen ratio (p10, l29) differs from the strategy of validation presented in Michel et al, 2016, for evaluating WACMOS-ET datasets. Is there a reason for that?

  **We would like to thank Dr. Ghilain for this comment. Because a lot of *in situ* datasets have no corrected measurements (due to missing variables necessary for the correction), the direct measurements were used in the end to validate the GLEAM datasets. This will be updated in the revised version of the manuscript. Note that in Michel *et al*. (2016), the datasets were validated against both the direct measurements and the energy balance residual. However, due to the lack of towers with measurements of all components of the surface energy balance, the latter approach was not considered here.**

- Extreme outliers are screened, what is the mask applied  For gaps in data, what is the strategy if less than 25.

  **Outliers are (subjectively) defined as measurements falling outside the following window, calculated for each time series separately: $[q_{25}-1.5(q_{75}-q_{25}); q_{75}+1.5(q_{75}-q_{25})]$, where $q_{75}$ and $q_{25}$ are the 75% and 25% quantiles of the dataset, respectively. Gaps in the remaining time series are not filled. This information will be added to the manuscript.**

---

## Referee Comment (RC1) · Anonymous Referee #1 · 30 Nov 2016

**OVERVIEW**

The manuscript describes the new version of the GLEAM dataset (v3) that includes land evaporation and root-zone soil moisture. The novelties in the retrieval algorithm and in the input datasets are firstly outlined. Secondly, three different datasets are generated by using different inputs and their quality is assessed through a comparison with in situ observations worldwide. Moreover, a cross-comparison with respect to the previous version of GLEAM (v2) is carried out.

**GENERAL COMMENTS**

The manuscript is well written and clear. The new release of the dataset is surely of interest for many research applications both in the hydrology and climate disciplines.

Moreover, the new release explicitly contains the root-zone soil moisture dataset that represent an additional benefit. The new release incorporates significant changes with respect to the previous version. Therefore, I believe the paper and the dataset deserve to be published on Geoscientific Model Development. Before the publication, I recognized some points that, in my opinion, need improvement and clarification.

1) MAJOR: In several Tables and Figures, the comparison between the three versions of v3 dataset, and against v2 dataset, is shown. In terms of soil moisture, it is highlighted that v3 performs better than v2 and that v03a is performing the best. However, I am wondering if the differences in the correlations between datasets are statistically significant. For instance, I believe that the differences reported in Table 3 for the overlap period between the three v3 datasets are not significant (median values between 0.61 and 0.65 for surface soil moisture). Therefore, I wouldn't stress too much that the new dataset is performing the best in terms of soil moisture, as the differences in the performance are quite small.

2) MODERATE: It is underlined several times that v03b and v03c are "fully satellite-based" datasets. It is not correct. The satellite rainfall product used as input is the gauge-corrected version of TMPA. As it is well-known, in TMPA 3B42v7 dataset ground observations are used for correcting the monthly totals month-by-month. They are not used for correcting the long-term bias, as it reads at lines 1-2, page 12). Therefore, in the gauge-corrected product the contribution of ground observations is significant (note that it could happen that the seasonal cycle is inverted between the real-time and the gauge-corrected version). I suggest removing the definition of these datasets as "fully satellite-based".

3) MODERATE: It would be interesting to show a version "d" of the v3 dataset in which SMOS observations are assimilated in the product using MSWEP as rainfall input. It would allow to disentangle the impact of rainfall forcing and the assimilated soil moisture product on the final quality of GLEAM datasets.

In the specific comments, I added some corrections and suggestions that should be implemented.

On this basis, I believe the paper deserves to be published only after a moderate revision.

**SPECIFIC COMMENTS (P: page, L: line or lines)**

P5, L2: I missed how snowmelt is computed. Can the authors add some details?

P6, L10: The paper by Lievens et al. (2016) is under review. As it is mentioned in the paper several times, and the readers do not have access to it, I believe some additional details should be included in this paper.

P6, L17-18: This sentence is also repeated below, I suggest removing.

P8, L12: I believe it should be specified that for w>wc S=1 and for w<wr S=0. Also for equation (5).

P10, L15-22: It is the third time in the paper that the three versions of v3 are described. Please try to avoid repetitions.

P11, L23-24: Why for a thicker model layer the representativeness of soil moisture measurements is lower? It should be explained.

P11, L32: Likely, it should be stressed also in the abstract that the quality assessment of root-zone soil moisture products is mostly carried out in CONUS region.

P12, L8-9: Strictly speaking, also the first model layer (10 cm) is thicker than the sensing depth of SMOS and ESA CCI soil moisture products. It should be acknowledged.

P13, L17: Figure 6 is not described in the text. Remove or add more details.

P13, L34: The possibility to correct for irrigation that is not modelled in GLEAM is highly interesting. However, it is not shown in the paper and, hence, the sentence should be
smoothed.

Figure 3: Specify explicitly which plot refers to tall and short vegetation.

[Figure]

---

## Author Comment (AC2) · 14 Dec 2016

**RC1**

**OVERVIEW**

The manuscript describes the new version of the GLEAM dataset (v3) that includes land evaporation and root-zone soil moisture. The novelties in the retrieval algorithm and in the input datasets are firstly outlined. Secondly, three different datasets are generated by using different inputs and their quality is assessed through a comparison with in situ observations worldwide. Moreover, a cross comparison with respect to the previous version of GLEAM (v2) is carried out.

**GENERAL COMMENTS**

The manuscript is well written and clear. The new release of the dataset is surely of interest for many research applications both in the hydrology and climate disciplines. Moreover, the new release explicitly contains the root-zone soil moisture dataset that represent an additional benefit. The new release incorporates significant changes with respect to the previous version. Therefore, I believe the paper and the dataset deserve to be published on Geoscientific Model Development. Before the publication, I recognized some points that, in my opinion, need improvement and clarification.

**We would like to thank the referee for reviewing the paper and giving some interesting comments and feedback. Below, we give a point-to-point reply to the comments posted by the reviewer.**

1. MAJOR: In several Tables and Figures, the comparison between the three versions of v3 dataset, and against v2 dataset, is shown. In terms of soil moisture, it is highlighted that v3 performs better than v2 and that v03a is performing the best. However, I am wondering if the differences in the correlations between datasets are statistically significant. For instance, I believe that the differences reported in Table 3 for the overlap period between the three v3 datasets are not significant (median values between 0.61 and 0.65 for surface soil moisture). Therefore, I wouldn't stress too much that the new dataset is performing the best in terms of soil moisture, as the differences in the performance are quite small.

   **We thank the referee for this comment and agree that we need to support the results with statistical significance tests. Therefore, in the revised version of the manuscript we will include the results of a statistical test to verify whether differences in correlations are significant or not. The discussion of the results will be based on these results as well.**

2. MODERATE: It is underlined several times that v03b and v03c are "fully satellite-based" datasets. It is not correct. The satellite rainfall product used as input is the gauge-corrected version of TMPA. As it is well-known, in TMPA 3B42v7 dataset ground observations are used for correcting the monthly totals month-by-month. They are not used for correcting the long-term bias, as it reads at lines 1-2, page 12). Therefore, in the gauge-corrected product the contribution of ground observations is significant (note that it could happen that the seasonal cycle is inverted between the real-time and the gauge-corrected version). I suggest removing the definition of these datasets as "fully satellite-based".

**We agree with the reviewer that the TMPA 3B42v7 product is not 'fully' satellite-based and that monthly totals are bias-corrected using gauge data. This is in clear contrast with the MSWEP dataset, where gauge-based products (e.g. the CPC-Unified dataset) are not used in a bias-correction step, but directly combined with other datasets using appropriate weights. This was indeed not clear from the original descriptions at P12-L1-4. We note however that the TMPA 3B42v7 product is described in the NASA website as a 'satellite precipitation product', and thus stating that the GLEAM output is solely based on satellite products as forcing is not – in our opinion – an overstatement. At the same time, we want to bring the reviewer's attention to the thin line separating what is a true satellite observation, since all the satellite observations used here have been calibrated using ground data after all... Nonetheless, and acknowledging the equivocality of this issue, we will avoid the use of 'fully' in the context of satellite-based forcing. We hope the reviewer can agree with this change.**

3. MODERATE: It would be interesting to show a version "d" of the v3 dataset in which SMOS observations are assimilated in the product using MSWEP as rainfall input. It would allow to disentangle the impact of rainfall forcing and the assimilated soil moisture product on the final quality of GLEAM datasets.
**This is indeed an interesting experiment, which we have already done in the past to confirm our results about the quality of the different input datasets and the performance of the assimilation algorithm. Replacing for instance the TMPA 3B42v7 for MSWEP in the v3.0c dataset increases the average open loop correlation (i.e. without data assimilation) of the first layer soil moisture against the in situ measurements from 0.61 to 0.66 (note that these statistics might be slightly different from the ones reported in the manuscript due to a reprocessing of the in situ data). This clearly indicates the higher quality of the MSWEP dataset in reference to the TMPA 3B42v7. If SMOS soil moisture observations are assimilated, both soil moisture datasets consistently improve over the CONUS, resulting in a slight increase of the same statistics to 0.62 and 0.67, respectively. These results indicate the high quality of the SMOS soil moisture dataset and the efficiency of the simple Newtonian Nudging algorithm.**

**Given that the paper is already quite extensive, including these results would not necessarily contribute to increasing the clarity of the manuscript. In addition, it is also not the main objective of the paper to focus on the quality of the different input datasets, neither to make strong claims about this. Therefore, the authors prefer not to include these results in the paper.**

In the specific comments, I added some corrections and suggestions that should be implemented. On this basis, I believe the paper deserves to be published only after a moderate revision.

**SPECIFIC COMMENTS (P: page, L: line or lines)**

1. P5, L2: I missed how snowmelt is computed. Can the authors add some details?

**This module of GLEAM is indeed not described in the paper. However, as this component of the model was not modified in reference to the original version, we would like to point readers to the first description of GLEAM in Miralles et al. (2011). A reference to the latter paper will be added to the revised version of the manuscript.**

2. P6, L10: The paper by Lievens et al. (2016) is under review. As it is mentioned in the paper several times, and the readers do not have access to it, I believe some additional details should be included in this paper.
   **The paper by Lievens et al. (2016) has recently been accepted and will be published online soon, so we will add the final citation to the revised manuscript.**

3. P6, L17-18: This sentence is also repeated below, I suggest removing.
   **Thanks. The second sentence will be removed.**

4. P8, L12: I believe it should be specified that for w>wc S=1 and for w<wr S=0. Also for equation (5).
   **We agree, yet this is already described in the original manuscript (P8-L1-4).**

5. P10, L15-22: It is the third time in the paper that the three versions of v3 are described. Please try to avoid repetitions.
   **This will be revised.**

6. P11, L23-24: Why for a thicker model layer the representativeness of soil moisture measurements is lower? It should be explained.
   **As the in situ soil moisture measurement is essentially a point measurement, it becomes less representative for the model if the volume to which it is compared gets larger (i.e. if the model layer gets thicker). When dealing with a 2D surface, the equivalent would be to think of the spatial representativeness of two different spatial resolutions (a coarse and a fine) and how they compare against a point measurement.**

7. P11, L32: Likely, it should be stressed also in the abstract that the quality assessment of root-zone soil moisture products is mostly carried out in CONUS region.
   **We agree with the reviewer and will include this information in the abstract.**

8. P12, L8-9: Strictly speaking, also the first model layer (10 cm) is thicker than the sensing depth of SMOS and ESA CCI soil moisture products. It should be acknowledged.
   **This is true and the resulting mismatch should be partly mitigated by the a priori bias removal. However, this is indeed not acknowledged in the paper and will be added in the revised version of the manuscript. We also acknowledge that the penetration depth of these sensors is variable, and can easily exceed 10 cm as well (see e.g.: *de Jeu, R.A.M and Holmes, T. Derivation of soil moisture sensing depth from microwave satellite sensors, Poster Presentation at the European Geosciences Union General Assembly 2015"*).**

9.  P13, L17: Figure 6 is not described in the text. Remove or add more details.
    **The results in Figure 6 were only briefly referred to at P13-L17-18 of the original paper. In the new manuscript, we will further elaborate on the results in Figure 6, but only briefly, since the conclusions are analogous to the ones that may be drawn from Figure 5.**

10. P13, L34: The possibility to correct for irrigation that is not modelled in GLEAM is highly interesting. However, it is not shown in the paper and, hence, the sentence should be smoothed.
    **The effects of irrigation on soil moisture should be partly captured by satellite-derived soil moisture datasets. As a result, a temporary increase in observed satellite soil moisture will likely result in an increase of the modelled soil moisture after data assimilation. However, since with the current validation data we are unable to detect this effect, we agree with the referee that this statement should be smoothed.**

11. Figure 3: Specify explicitly which plot refers to tall and short vegetation.
    **We would like to emphasize that the same stress function for short and tall vegetation is implemented in GLEAM v3. Therefore, the panels in Figure 3 do not necessarily refer to either short or tall vegetation, but rather show the effect of the VOD on the stress (a large range in VOD vs. a small range in VOD). We will update the caption in Figure 3 to make this clear.**

---

## Referee Comment (RC2) · Anonymous Referee #2 · 14 Feb 2017

This is a well-written manuscript whose conclusions are derived via a methodologically evaluation analysis. I think that the (clear) description and evaluation of the version-to-version differences constitutes a significant enough contribution to warrant publication. However – prior to that point – the following two major points should be addressed.

Major points:

1) In the introduction, the author's argue that GLEAM is unique in that it is "primarily driven by microwave remote sensing observations." So the novelty here really seems to spring from 1) the assimilation of microwave-based soil moisture and 2) the use of microwave-based vegetation optical depth in the canopy stress formulation. If you take away these two aspects, the approach really just collapses down into a basic rain-driven soil water balance approach (which is relatively simple compared to the combined water/energy balance land surface already being run globally in e.g. GLDAS).

So it would strengthen the paper if there were more support for the assertion that GLEAM is driven "primarily" by surface microwave observations.

Figure 5 and 6 are clearly an attempt to do this. . .but the results are not very compelling. The second and third columns of Figure 5 show that the background water balance model is generally superior to the assimilated observations. So naturally, more weight is (generally) placed on the water balance model background. This is ok. . .but it is really consistent with GLEAM being "primarily" driven by the microwave surface observations? Instead, it seems more accurate to say that GLEAM is being "primarily" driven by water balance considerations and these balance considerations are being nudged by "secondary" considerations derived from microwave DA.

No comparable results are shown for either root-zone soil moisture or ET. . .presumably because the impact of microwave DA is even less for these outputs.

I realize that some of this is just semantics (i.e. what constitutes "primary" versus "secondary"). . .but I do think that the authors should either: 1) present better evidence for the "primary" role of the microwave observations in GLEAM or 2) be more objective in describing the novelty of their approach. . .particularly the impact of their novel methodological elements relative to approaches (like a classical soil water balance model) which have been around for quite some time.

2) Some type of statistical significance analysis is needed to assess the noted version-to-version differences. I do not think that "statistically-significant" differences should be a requirement for publication. Nevertheless, the reader should be given a sense as to how large the stated performance differences are relative to expected levels of sampling noise.

Minor points:

1) Page 1, Line 7. . .I'd stay away from subjective statements like "most of these variables can be relatively easily observed at different spatial scales"...it is a stretch to call the remote estimation of rainfall (for example) "easy"...much safer to say from the remote retrieval of ET is difficult relative to other water balance components.

2) Figure 6 does not seem to be references in the manuscript. Also, unclear why case 3c is dropped when moving from Fig. 5 to Fig. 6.

---

## Author Comment (AC3) · 15 Feb 2017

**RC2**

**OVERVIEW**

This is a well-written manuscript whose conclusions are derived via a methodologically evaluation analysis. I think that the (clear) description and evaluation of the version-to-version differences constitutes a significant enough contribution to warrant publication. However – prior to that point – the following two major points should be addressed.

**We thank the referee for assessing the quality of the paper and giving some interesting comments. Below, we give a point-to-point reply to the comments posted by the reviewer.**

**MAJOR COMMENTS**

1. In the introduction, the author's argue that GLEAM is unique in that it is "primarily driven by microwave remote sensing observations." So the novelty here really seems to spring from 1) the assimilation of microwave-based soil moisture and 2) the use of microwave-based vegetation optical depth in the canopy stress formulation. If you take away these two aspects, the approach really just collapses down into a basic rain-driven soil water balance approach (which is relatively simple compared to the combined water/energy balance land surface already being run globally in e.g. GLDAS).

   **We thank the referee for this comment. We would like to emphasize that (a) not only the soil moisture and vegetation optical depth datasets are based on microwave observations, but that the entire dynamic forcing dataset of the GLEAM v3b and c are based on satellite observations, and (b) a third important feature of the model is the detailed estimation of interception loss via the modified Gash's analytical model (Miralles et al., 2010), which was (for instance) used to benchmark the MERRA reanalysis and correct its interception estimates in new releases (Reichle at al., 2017). Therefore, we claim that the model can be primarily driven by satellite observations, being in their vast majority of microwave nature (soil moisture, precipitation, vegetation optical depth); thus available also during cloudy conditions, which is a unique feature for this type of models dedicated to estimate terrestrial evaporation from remotely-sensed data, since other models (such as e.g. Zhang et al., 2010; Fisher et al., 2008; Mu et al., 2007) need to rely on reanalysis meteorology, due to the requirements of atmospheric humidity, and on optical greenness data.**

   **We fully agree with the referee that the approach is simpler than land-surface models such as GLDAS, which provide a more complete representation of land surface processes: the added value of GLEAM is that it is specifically designed to estimate terrestrial evaporation and that it has been thoroughly evaluated and validated in regards to its skill to perform this very specific task. Needless to say that the variability in the representation of evaporation in more complex models is actually very large (Jimenez et al., 2011), mostly due to the fact that these models**

**are not specifically developed to estimate the evaporation flux accurately, independently of their complexity. Nonetheless, it should be noted that it is not our intention to present these features of GLEAM as innovative, as the core of the model was developed in 2011 based on this same rationale. We will do an effort to incorporate these points in the revised version.**

So it would strengthen the paper if there were more support for the assertion that GLEAM is driven "primarily" by surface microwave observations.

Figure 5 and 6 are clearly an attempt to do this… but the results are not very compelling. The second and third columns of Figure 5 show that the background water balance model is generally superior to the assimilated observations. So naturally, more weight is (generally) placed on the water balance model background. This is ok… but it is really consistent with GLEAM being "primarily" driven by the microwave surface observations? Instead, it seems more accurate to say that GLEAM is being "primarily" driven by water balance considerations and these balance considerations are being nudged by "secondary" considerations derived from microwave DA.

**We would first like to stress that the water balance is primarily driven by microwave-based precipitation, and evaporation estimates based on microwave precipitation, microwave vegetation optical depth and satellite-based (or reanalysis) meteorology. We agree nonetheless that the text should state that GLEAM is mostly driven by satellite data, which are primarily derived from microwave sensors. Further, Figures 5 and 6 mainly show that the impact of the DA on the modelled surface soil moisture strongly depends on the quality of the model open loop soil moisture, which on his turn is highly impacted by the quality of the precipitation forcing. We will clarify these points further in the manuscript.**

No comparable results are shown for either root-zone soil moisture or ET… presumably because the impact of microwave DA is even less for these outputs.

**The impact of the data assimilation system on the estimated evaporation is indeed limited and not discussed here. For a more detailed discussion about the impact of the data assimilation on evaporation, we would like to point the referee to Martens et al. (2016).**

I realize that some of this is just semantics (i.e. what constitutes "primary" versus "secondary")… but I do think that the authors should either: 1) present better evidence for the "primary" role of the microwave observations in GLEAM or 2) be more objective in describing the novelty of their approach… particularly the impact of their novel methodological elements relative to approaches (like a classical soil water balance model) which have been around for quite some time.

**We note again that the precipitation is also microwave-based (to the largest extent). In the revised version of the manuscript, we will try to be more specific about what we can consider novel features of our approach (see first response).**

2. Some type of statistical significance analysis is needed to assess the noted version-to-version differences. I do not think that "statistically-significant" differences should be a requirement for publication. Nevertheless, the reader should be given a sense as to how large the stated performance differences are relative to expected levels of sampling noise.

**We agree with the reviewer that we need to support the results with statistical significance tests. Therefore, in the revised version of the manuscript we will include the results of a statistical test to verify whether differences in correlations are significant or not. The discussion of the results will be based on these tests as well. We note, nevertheless, that the two versions are similar on their estimates, and that the rationale for updating the method has been to make it more realistic while keeping the simplicity of the algorithm, as well as extending substantially the dataset temporal record based on the adoption of a new range of forcing data.**

**MINOR COMMENTS**

1. Page 1, Line 7… I'd stay away from subjective statements like "most of these variables can be relatively easily observed at different spatial scales"… it is a stretch to call the remote estimation of rainfall (for example) "easy"… much safer to say from the remote retrieval of ET is difficult relative to other water balance components.

   **This will be updated.**

2. Figure 6 does not seem to be references in the manuscript. Also, unclear why case 3c is dropped when moving from Fig. 5 to Fig. 6.

   **The results in Figure 6 are only briefly referred to at P13-L17-18 of the original paper, since the conclusions are analogous to the ones that may be drawn from Figure 5. In the new manuscript, we will further elaborate on the results in Figure 6. Anomaly correlations are not calculated for the GLEAMv3.0c (and thus not shown) as the period covered by this product is only 5 years (2011–2015). Given that none of the _in situ_ stations fully covers this period with measurements, it is believed that this period is too short to calculate a robust climatology. This will be mentioned in the revised manuscript.**

---

## Author Response (AR1)

Dear GMD editor,

First, we would like to thank the associate editor and reviewers for handling our manuscript. Enclosed to this letter, a revised version of our manuscript GMD-2016-162 can be found. We believe we have addressed all comments raised by the reviewers and modified the manuscript accordingly. The most important changes can be found in Sect. 4, where the results of statistical significance tests are added and discussed. In addition, some textual issues have been fixed and the information from the original Tables 1 and 2 has been merged. Finally, we would also like to highlight that the v3a GLEAM dataset has been extended until 2015, and that the in situ validation data has been updated to their latest versions. Although this only resulted in minor changes in the validation statistics, the relevant figures have been updated accordingly. Below, a list of the comments per reviewer is provided, together with our reply and how we modified things in the manuscript. We hope that this revised version of the manuscript is eligible for publication in GMD.

Brecht Martens,

On behalf of all co-authors.

RC1 (published online: 30/11/2016):

1. MAJOR: In several Tables and Figures, the comparison between the three versions of v3 dataset, and against v2 dataset, is shown. In terms of soil moisture, it is highlighted that v3 performs better than v2 and that v03a is performing the best. However, I am wondering if the differences in the correlations between datasets are statistically significant. For instance, I believe that the differences reported in Table 3 for the overlap period between the three v3 datasets are not significant (median values between 0.61 and 0.65 for surface soil moisture). Therefore, I wouldn't stress too much that the new dataset is performing the best in terms of soil moisture, as the differences in the performance are quite small.

   We thank the referee for this comment and agree that we need to support the results with statistical significance tests. Therefore, in the revised version of the manuscript we have included the results from a statistical test to verify whether differences in correlations are significant or not.

   *Changes in manuscript*
   Statistical significance tests have been performed to analyse the differences in correlations against in situ measurements between different datasets and/or experiments. The test used here is described at P11-L8-11. The results of these tests are discussed throughout Sect. 4: 'Results and Discussion'.

2. MODERATE: It is underlined several times that v03b and v03c are 'fully satellite-based' datasets. It is not correct. The satellite rainfall product used as input is the gauge-corrected version of TMPA. As it is well-known, in TMPA 3B42v7 dataset ground observations are used for correcting the monthly totals month-by-month. They are not used for correcting the long-term bias, as it reads at lines 1-2, page 12). Therefore, in the gauge-corrected product the contribution of ground observations is significant (note that it could happen that the seasonal cycle is inverted between the real-time and the gauge-corrected version). I suggest removing the definition of these datasets as 'fully satellite-based'.

   We agree with the reviewer that the TMPA 3B42v7 product is not 'fully' satellite-based and that monthly totals are bias-corrected using gauge data (if gauges are in close proximity). This is in clear contrast with the MSWEP dataset, where gauge-based products (e.g. the CPC-Unified dataset) are not used in a bias-correction step, but directly combined with other datasets using appropriate weights. We note however that the TMPA 3B42v7 product is described in the NASA website as a 'satellite precipitation product'. We have opted therefore to avoid referring to GLEAM v3b and c as 'solely based

on satellite data', and referred to the fact that 'they are driven by satellite-based data only'. We hope the reviewer can agree with this change.

*Changes in manuscript*
The data set descriptions have been updated and the use of 'solely based on satellite data' or 'fully satellite-based' have been avoided throughout the manuscript. In addition, at P8-L19-21 we clearly acknowledge now that the TMPA dataset is bias-corrected at the monthly scale using gauge data.

3. MODERATE: It would be interesting to show a version 'd' of the v3 dataset in which SMOS observations are assimilated in the product using MSWEP as rainfall input. It would allow to disentangle the impact of rainfall forcing and the assimilated soil moisture product on the final quality of GLEAM datasets.

This is indeed an interesting experiment, which we have already done in the past to confirm our results about the quality of the different input datasets and the performance of the assimilation. Replacing for instance the TMPA 3B42v7 for MSWEP in the v3.0c dataset increases the average open loop correlation (i.e. without data assimilation) of the first layer soil moisture against the in situ measurements from 0.61 to 0.66. This clearly reflects the higher quality of the MSWEP dataset in reference to the TMPA 3B42v7. If SMOS soil moisture observations are assimilated, both soil moisture datasets consistently improve over the CONUS, resulting in a slight increase of the same statistics to 0.62 and 0.67, respectively. These results highlight the high quality of the SMOS soil moisture dataset and the efficiency of the simple Newtonian Nudging algorithm.

*Changes in manuscript*
Given that the paper is already quite extensive, we believe that including these results would not contribute to increasing the clarity of the manuscript. However, at P11-L24-26 a brief note about the above-mentioned experiments can be found.

4. P5, L2: I missed how snowmelt is computed. Can the authors add some details?

This module of GLEAM is indeed not described in the paper. However, as this component of the model was not modified in reference to the original version, we would like to point readers to the first description of GLEAM in Miralles et al. (2011).

*Changes in manuscript*
A reference to the latter has been added at P4-L18-19 of the revised manuscript. In addition, at P4-L23-25, readers are pointed to the original papers for more detailed descriptions of the model baseline.

5. P6, L10: The paper by Lievens et al. (2016) is under review. As it is mentioned in the paper several times, and the readers do not have access to it, I believe some additional details should be included in this paper.

The paper by Lievens et al. (2017) has recently been accepted and has been published now.

*Changes in manuscript*
An updated citation for Lievens et al. (2017) has been added to the revised manuscript.

6. P6, L17-18: This sentence is also repeated below, I suggest removing.

We thank the referee for this comment.

*Changes in manuscript*
The sentence at P6-L26-27 of the original paper has been removed.

7. P8, L12: I believe it should be specified that for w>wc S=1 and for w<wr S=0. Also for equation (5).

We agree, yet this was already described in the original manuscript (P8-L1-4).

*Changes in manuscript*
As this is already described in the paper, no changes have been made.

8. P10, L15-22: It is the third time in the paper that the three versions of v3 are described. Please try to avoid repetitions.

We agree with the reviewer on this point.

*Changes in manuscript*
The description of the three datasets in the introduction has been removed (P3-L16-24 of the original manuscript).

9. P11, L23-24: Why for a thicker model layer the representativeness of soil moisture measurements is lower? It should be explained.

As the in situ soil moisture measurement is essentially a point measurement, it becomes less representative for the model if the volume to which it is compared gets larger (i.e. if the model layer is thicker). When dealing with a 2D surface, the equivalent would be to think of the spatial representativeness of two different spatial resolutions (a coarse and a fine) and how they compare against a point measurement.

*Changes in manuscript*
This information has not been added to the manuscript, as we believe this is clear from the context of the statement.

10. P11, L32: Likely, it should be stressed also in the abstract that the quality assessment of root-zone soil moisture products is mostly carried out in CONUS region.

We believe this is already well highlighted in the manuscript (e.g. P11-L26-27, P12-L3-4, P12-L20 etc.). In addition, we did an effort to make the validation as global as possible, and also note that several hundred of soil moisture sites (approximately 500) are located outside the CONUS.

*Changes in manuscript*
This information has not been added to the abstract of the manuscript.

11. P12, L8-9: Strictly speaking, also the first model layer (10 cm) is thicker than the sensing depth of SMOS and ESA CCI soil moisture products. It should be acknowledged.

This is true and the resulting mismatch should be partly mitigated by the bias removal. However, this is indeed not acknowledged in the paper and has been added to the revised version of the manuscript. We also acknowledge that the penetration depth of these sensors is variable, and can easily exceed 10 cm as well (see e.g.: *'de Jeu, R.A.M and Holmes, T. Derivation of soil moisture sensing depth from microwave satellite sensors, Poster Presentation at the European Geosciences Union General Assembly 2015'*).

*Changes in manuscript*

It has been acknowledged at P5-L28-29 of the revised version of the manuscript that microwave-based soil moisture datasets are typically only representative of the first few centimetres of the soil.

12. P13, L17: Figure 6 is not described in the text. Remove or add more details.

The results in Figure 6 were referred to at P13-L17-18 of the original paper.

*Changes in manuscript*
As the conclusions drawn from Figure 6 are analogous to the ones that may be drawn from Figure 5, we have decided not to further elaborate on these results.

13. P13, L34: The possibility to correct for irrigation that is not modelled in GLEAM is highly interesting. However, it is not shown in the paper and, hence, the sentence should be smoothed.

The effects of irrigation on soil moisture should be partly captured by satellite-derived soil moisture datasets. As a result, a temporary increase in observed satellite soil moisture will likely result in an increase of the modelled soil moisture after data assimilation. However, since with the current validation data we are unable to detect this effect, we agree with the referee that this statement should be softened.

*Changes in manuscript*
We have smoothed this sentence by adding the word 'assumed' at P13-L25-27: 'Nevertheless, our simple Newtonian Nudging data assimilation system is still assumed to correct for random forcing errors, and potential other effects such as irrigation, that are not explicitly modelled in GLEAM.'

14. Figure 3: Specify explicitly which plot refers to tall and short vegetation.

We would like to emphasize that the same stress function for short and tall vegetation is implemented in GLEAM v3. Therefore, the panels in Figure 3 do not necessarily refer to either short or tall vegetation, but rather show the effect of the VOD on the stress (a large range in VOD vs. a small range in VOD).

*Changes in manuscript*
To make things clearer, the caption of Figure 3 has been slightly updated. In addition, the sentence at P7-L12-14 has been modified to make clear that there is only one stress function (no plural).

RC2 (published online: 14/02/2016):

1. In the introduction, the author's argue that GLEAM is unique in that it is 'primarily driven by microwave remote sensing observations.' So the novelty here really seems to spring from 1) the assimilation of microwave-based soil moisture and 2) the use of microwave-based vegetation optical depth in the canopy stress formulation. If you take away these two aspects, the approach really just collapses down into a basic rain-driven soil water balance approach (which is relatively simple compared to the combined water/energy balance land surface already being run globally in e.g. GLDAS).

We thank the referee for this comment. We would like to emphasize that (a) not only the soil moisture and vegetation optical depth datasets are based on satellite observations, but that the entire dynamic forcing dataset of the GLEAM v3b and c is based on satellite observations, and (b) another important feature of the model is the detailed estimation of interception loss via the modified Gash's analytical model (Miralles et al., 2010), which was (for instance) used to benchmark the MERRA reanalysis and correct its interception estimates in new releases (Reichle at al., 2017). Therefore, we claim that the model can be primarily driven by satellite observations, being in their vast majority of microwave

nature (soil moisture, precipitation, vegetation optical depth); thus available also during cloudy conditions, which is unique for this type of models dedicated to estimate terrestrial evaporation from remotely-sensed data, since other models (such as e.g. Zhang et al., 2010; Fisher et al., 2008; Mu et al., 2007) are forced to rely on reanalysis meteorology due to the requirements of atmospheric humidity and/or wind speed, and on optical greenness data.

In addition, we fully agree with the referee that the approach is simpler than land-surface models such as GLDAS, which provide a more detailed representation of land processes. The added value of GLEAM is that it is specifically designed to estimate terrestrial evaporation and that it has been thoroughly evaluated and validated in regards to its skill to perform this very specific task. Needless to say that the uncertainty in the representation of evaporation in more complex models is actually very large (Jiménez et al., 2011), mostly due to the fact that these models are not specifically developed to estimate the evaporation flux accurately. Nonetheless, it should be noted that it is not our intention to present these features of GLEAM as innovative, as the core of the model was developed in 2011 based on this same rationale. Consequently, we have done an effort to incorporate these points in the revised version of the manuscript (see below).

So it would strengthen the paper if there were more support for the assertion that GLEAM is driven 'primarily' by surface microwave observations. Figure 5 and 6 are clearly an attempt to do this... but the results are not very compelling. The second and third columns of Figure 5 show that the background water balance model is generally superior to the assimilated observations. So naturally, more weight is (generally) placed on the water balance model background. This is ok... but it is really consistent with GLEAM being 'primarily' driven by the microwave surface observations? Instead, it seems more accurate to say that GLEAM is being 'primarily' driven by water balance considerations and these balance considerations are being nudged by 'secondary' considerations derived from microwave DA.

We would first like to stress that the water balance is primarily driven by microwave-based precipitation, and evaporation estimates based on microwave-based precipitation, microwave-based vegetation optical depth and satellite-based (or reanalysis) meteorology. In addition, microwave-based soil moisture is assimilated as stated by the referee. We agree nonetheless that the text should state clearly that GLEAM is mostly 'driven by satellite data', which are 'primarily derived from microwave sensors'. While Figures 5 and 6 mainly show that the impact of the DA on the modelled surface soil moisture strongly depends on the quality of the model open loop soil moisture, the latter is highly impacted by the quality of the precipitation forcing which is largely microwave-based as well.

No comparable results are shown for either root-zone soil moisture or ET... presumably because the impact of microwave DA is even less for these outputs.

The impact of the data assimilation system on the estimated evaporation is indeed limited and not discussed here. For a more detailed discussion about the impact of the data assimilation on evaporation, we would like to point the referee to the study by Martens et al. (2016).

I realize that some of this is just semantics (i.e. what constitutes 'primary' versus 'secondary')... but I do think that the authors should either: 1) present better evidence for the 'primary' role of the microwave observations in GLEAM or 2) be more objective in describing the novelty of their approach... particularly the impact of their novel methodological elements relative to approaches (like a classical soil water balance model) which have been around for quite some time.

We agree, yet we note again that the precipitation is also microwave-based (to the largest extent). In the revised version of the manuscript.

*Changes in manuscript*

We have tried to clarify the points raised by the reviewer throughout the manuscript: (a) we do not claim anymore that GLEAM is only driven by microwave remote sensing observations, and (b) we have tried to clarify which aspects of the model are novel or not. These changes can be found throughout the manuscript. For instance, at P2-L32-33 we clearly state that the model is designed 'to be driven' by remote sensing observations, which are 'primarily' derived from microwave data (not only). Around the same lines, we also list the two key features of the model, and at P3-L7-14 and P15-L27-29, we clearly list the novel aspects/new algorithms implemented in GLEAM.

2. Some type of statistical significance analysis is needed to assess the noted version-to-version differences. I do not think that 'statistically-significant' differences should be a requirement for publication. Nevertheless, the reader should be given a sense as to how large the stated performance differences are relative to expected levels of sampling noise.

We agree with the reviewer that we need to support the results with statistical significance tests. Therefore, in the revision, we have included tests to verify whether differences in correlations are significant. The discussion of the results throughout Sect. 4: 'Results and Discussion' incorporates the results of these texts. We note, nevertheless, that the two versions are similar on their estimates, and that the rationale for updating the method has been to make it more physically realistic while keeping the simplicity of the algorithm, as well as extending substantially the dataset temporal record based on the adoption of a new range of forcing data.

*Changes in manuscript*
Statistical significance tests have been performed to analyse the differences in correlation against in situ measurements between different datasets and/or experiments. The test used here is described at P11-L8-11. The results of these tests are discussed throughout Sect. 4.

3. Page 1, Line 7... I'd stay away from subjective statements like 'most of these variables can be relatively easily observed at different spatial scales'... it is a stretch to call the remote estimation of rainfall (for example) 'easy'... much safer to say from the remote retrieval of ET is difficult relative to other water balance components.

*Changes in manuscript*
The reference to the observability of the other hydrological variables has been removed and this sentence has been changed to: 'Unfortunately, the large-scale observation of terrestrial evaporation is hampered by the inability to sense this flux directly from satellites.' (P2-L7-8).

4. Figure 6 does not seem to be references in the manuscript. Also, unclear why case 3c is dropped when moving from Fig. 5 to Fig. 6.

The results in Figure 6 were only briefly referred to at P13-L17-18 of the original paper, since the conclusions were analogous to the ones that may be drawn from Figure 5. Anomaly correlations are not calculated for the GLEAMv3c (and thus not shown) as the period covered by this product is only 5 years (2011–2015). Given that none of the *in situ* stations fully covers this period with measurements (without any data gaps), it is believed that this period is too short to calculate a robust climatology.

*Changes in manuscript*
As the conclusions drawn from Figure 6 are analogous to the ones that may be drawn from Figure 5, we have decided not to further elaborate on the results related to this figure.
Regarding the anomaly correlations, the justification for not including the v3c dataset in this analysis, has been added to the manuscript at P12-L10-12.

[revised manuscript text omitted]